# Bioinformatics analysis of the potentially functional circRNA-miRNA-mRNA network in breast cancer

Cihat Erdogan[1], Ilknur Suer[2,3], Murat Kaya[3], Sukru Ozturk[3], Nizamettin Aydin[4], Zeyneb Kurt[5]*

1 Department of Medical and Molecular Genetics, Indiana University School of Medicine, Indianapolis, Indiana, United States of America, 2 Department of Medical Genetics, Istanbul Faculty of Medicine, Istanbul University, Istanbul, Turkey, 3 Department of Internal Medicine, Division of Medical Genetics, Istanbul Faculty of Medicine, Istanbul University, Istanbul, Turkey, 4 Department of Computer Engineering, Faculty of Computer and Informatics, Istanbul Technical University, Istanbul, Turkey, 5 Information School, The University of Sheffield, Sheffield, United Kingdom

☺ These authors contributed equally to this work.
* z.kurt@sheffield.ac.uk

**Data Availability Statement:** All circRNA expression profiles are available from the Gene Expression Omnibus (GEO) database (accession numbers GSE101124 and GSE182471). All miRNA

## Abstract

Breast cancer (BC) is the most common cancer among women with high morbidity and mortality. Therefore, new research is still needed for biomarker detection. GSE101124 and GSE182471 datasets were obtained from the Gene Expression Omnibus (GEO) database to evaluate differentially expressed circular RNAs (circRNAs). The Cancer Genome Atlas (TCGA) and Molecular Taxonomy of Breast Cancer International Consortium (METABRIC) databases were used to identify the significantly dysregulated microRNAs (miRNAs) and genes considering the Prediction Analysis of Microarray classification (PAM50). The circRNA-miRNA-mRNA relationship was investigated using the Cancer-Specific CircRNA, miRDB, miRTarBase, and miRWalk databases. The circRNA–miRNA–mRNA regulatory network was annotated using Gene Ontology (GO) analysis and Kyoto Encyclopedia of Genes and Genomes (KEGG) pathway database. The protein-protein interaction network was constructed by the STRING database and visualized by the Cytoscape tool. Then, raw miRNA data and genes were filtered using some selection criteria according to a specific expression level in PAM50 subgroups. A bottleneck method was utilized to obtain highly interacted hub genes using cytoHubba Cytoscape plugin. The Disease-Free Survival and Overall Survival analysis were performed for these hub genes, which are detected within the miRNA and circRNA axis in our study. We identified three circRNAs, three miRNAs, and eighteen candidate target genes that may play an important role in BC. In addition, it has been determined that these molecules can be useful in the classification of BC, especially in determining the basal-like breast cancer (BLBC) subtype. We conclude that hsa_-circ_0000515/miR-486-5p/SDC1 axis may be an important biomarker candidate in distinguishing patients in the BLBC subgroup of BC.

and mRNA expression data as well as meta-data of the samples are available on the TCGA database (the breast cancer cohort on TCGA).

**Funding:** The author(s) received no specific funding for this work.

**Competing interests:** The authors have declared that no competing interests exist.

# 1 Introduction

Breast cancer (BC) is a heterogeneous type of malignancy that occurs as a result of distinct molecular alterations in breast tissue [1]. Circular RNAs (circRNAs) are evolutionarily conserved and stable RNA regulators that can behave as microRNA (miRNA) sponges, regulate alternative splicing mechanisms, and take an active role in the expression of the gene in which they are encoded [2]. circRNAs have been shown to play crucial roles in the cell, and in recent years this RNA class has been one of the most important research focuses, particularly in the field of cancer [3]. It has been noted that biological processes in living cells would better be modeled by networks since molecular phenotypes do not operate in isolation, instead their interactions collectively carry out these processes [4]. Hence, a network representation can provide a better understanding of the biological and molecular processes beyond analyzing a single molecule or gene, for instance identified from differentially expressed gene analyses. It is expected that identifying circRNA-miRNA-mRNA connections will be essential in explaining the molecular processes of numerous illnesses, detecting biomarkers for early diagnosis, and expanding therapy choices. A single miRNA has the capacity to target hundreds of genes, while a single circRNA can serve as a sponge for dozens of miRNAs. Using bioinformatics data to simplify the circRNA-miRNA-mRNA interactions, which are comprised of such complicated processes, can shed light on in vitro and in vivo investigations. For example, in the bioinformatics study of Liu et al [5], it was emphasized that *hsa_circRNA_0003638* may play a role in the pathogenesis of atrial fibrillation by targeting the *CXCR4* gene via hsa-miR-1207-3p. Similarly, Hu et al [6] suggested that the interaction of *hsa_circ_0009581/hsa-miR-150-5p*, and *hsa_circ_0001947/hsa-miR-454-3p* may play a role in the AML cancer process. The precise biological classification of the BC subtype is critical for predicting the disease's progression. Clinical management of BC is dependent on criteria such as tumor size, age, Estrogen (ER) and Progesterone (PR) expression, and the presence or absence of amplification and concurrent enhanced Human epidermal growth factor receptor 2 (HER2) expression. However, these indicators are currently insufficient for accurately categorizing individuals into sections with a high or low risk of relapse, as well as identifying subgroups resistant to therapy [7]. Technological breakthroughs in recent decades have enabled molecular classification based on distinct global gene expression. mRNA expression patterns assessed using microarrays revealed that BC had distinct intrinsic fingerprints that may be utilized to classify tumors into intrinsic molecular subgroups [8–11]. Despite considerable advances in this field, there is still a need for novel markers to refine categorization, particularly for some subtypes [7]. Studies on circRNA, which is a relatively new field of research area, and its relationship with BC subtypes are still quite insufficient. The identification of new genes with variable expression across different types of BC, as well as the detection of miRNAs and circRNAs associated with these genes, may be critical for cancer categorization and potential treatment. Therefore, in our study we demonstrated the circRNA-miRNA-mRNA regulatory connections in BC subtypes using various databases (S1 Table). The circRNAs were detected using the GSE101124 [12] and GSE182471 [13] datasets. miRNAs with significantly altered expression in Prediction Analysis of Microarray (PAM50) subtypes were identified using The Cancer Genome Atlas (TCGA) and Molecular Taxonomy of Breast Cancer International Consortium (METABRIC) datasets. The TCGA dataset was also used to identify genes with dramatically changed expressions in BC. The circRNA-miRNA-mRNA relationship was investigated for each PAM50 subtype using the Cancer-Specific CircRNA (*CSCD*) [14], *miRDB* [15], *miRWalk* [16], and *miRTarBase* [17] databases. Previously, overall BC-associated circRNA-miRNA-mRNA interactions have been found out [18] but this was not investigated for different PAM50 subtypes individually. Also, solely a single dataset for each molecular data type has been used previously.

We identified those interactions for each PAM50 subtype using multiple datasets per data type. Although the PAM50 subtyping was available for the miRNA and gene expression data-sets, this information does not exist for the circRNA datasets. We have initiated our examination from the upper stream of the candidate circRNA-miRNA-mRNA axes and the PAM50 subtype-associated shortlisted molecules were mapped to the downstream until the subtype information was not available any further. We have proceeded our further investigation and in-silico confirmation for the differentially regulated miRNAs and genes that are shared across all PAM50 subtypes with a further emphasize on implications of our findings on the basal-like breast cancer (BLBC) molecular subtype since it is the most aggressive one among all with a higher recurrence rate and poorer outcome for 5-year survival [19].

## 2 Materials and methods

### 2.1 Differentially expression analysis of BC datasets

The circRNA, miRNA and mRNA expressions were analyzed using various databases and datasets. The block diagram of our pipeline is illustrated in Fig 1.

**2.1.1 circRNA expression.** The circRNA expression profiles were gathered from the Gene Expression Omnibus (GEO) database with an access code of GSE101124 [12] (the dataset includes four BC cell samples, four triple negative breast cancer (TNBC) and four Luminal-A (LumA) molecular subtype tissue samples and three non-tumoruos mammary gland tissue samples) and GSE182471 [13] (the dataset includes five BC samples and five non-tumor samples). Four cancer cell lines in the GSE101124 dataset have been removed and the eight tumor-ous and three non-tumorous tissue samples have been kept. Molecular subtypes of the samples in the dataset GSE182471 were not annotated. Then, the differentially expressed circRNAs (DECs) were identified by using the *limma* R package (v.3.46.0) with a p-value less than 0.05 and an absolute log2-transformed FC (fold change) value of $\geq 1$.

**2.1.2 miRNA and mRNA expression.** The miRNA and mRNA expression data as well as meta-data of the samples were downloaded from the TCGA database [20]. The TCGA dataset contains alterations in miRNA and mRNA expressions from 901 BC samples (162 basal-like, 73 HER2-positive, 455 Luminal A, 178 Luminal B, 33 normal-like) and 112 control samples.

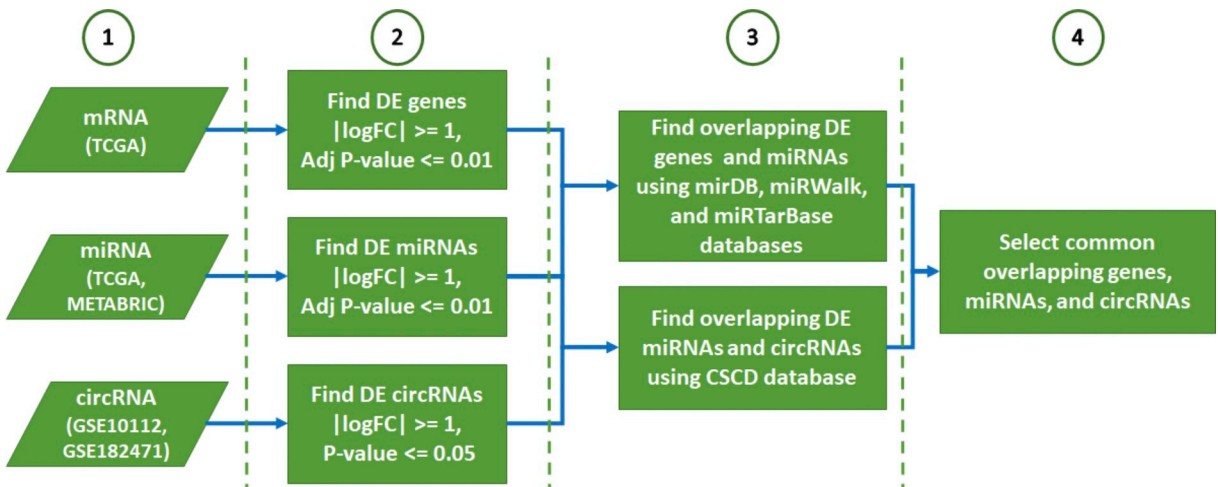

**Fig 1. The steps of the BC PAM50 subtype analysis.** The miRNA–mRNA interactions were estimated with *mirDB* (v6), *miRTarBase* (Release 8.0), and *miRWalk* (v3). DE: Differentially expressed, *CSCD (v2.0)*: The Cancer-specific circRNAs database, BC: Breast cancer, FC: Fold change, log: logarithm base 2.

Also, the METABRIC (University of Cambridge) dataset [21] was obtained from the European Genome-Phenome Archive (EGA-S00000000122) to validate our findings. The METABRIC dataset includes 1,301 BC samples (198 basal-like, 161 HER2-positive, 461 Luminal A, 370 Luminal B, 99 normal-like, and 12 unknown tumors) and 116 control samples of the miRNA expression analyses. We used the *DESeq2* R package (v.1.28.1) to determine differentially expressed mRNAs (DEGs) and miRNAs (DEMs) with a set of criteria, False Discovery Rate (FDR) < 0.01 and an absolute log2FC value of ≥1 (for both mRNAs and miRNAs), for each group of the PAM50 classification across the TCGA samples in comparison to the healthy tissue samples. Since there are no control samples in the mRNA expression dataset from the METABRIC database, we conducted the DEG analysis only on the TCGA.

Since the raw data was not provided, but only the normalized data were available in the METABRIC database, the DEMs were determined with the same criteria given above by using the *limma* R package. Overlapping mRNA and miRNAs from differential expression analyses were curated for further analysis. All FDR values were obtained by the Benjamini-Hochberg method.

## 2.2 Predicting the associated biological features

In order to predict circRNAs and miRNAs interactions, the most significantly altered 13 DECs (both down- and up-regulated) were chosen via the *CSCD* v2.0 database. On the other hand, the *miRDB*, *miRWalk*, and *miRTarBase* databases were used to find the interactions between the DEGs and DEMs across all of the PAM50 classes. Hence, the knowledge base-driven target mRNAs of the up-regulated miRNAs in all PAM50 classes were searched among the down-regulated mRNAs, whereas the down-regulated miRNAs' targets were searched among the up-regulated mRNAs. Similarly, knowledge base-driven target miRNAs of the up-regulated circRNAs were searched among the down-regulated miRNAs, whereas the targets of the down-regulated circRNAs were searched among the up-regulated miRNAs. After that, the interactions of circRNAs, miRNAs and mRNAs, which were found to have the most significant expression change, were investigated from the literature.

## 2.3 Selection criteria for filtered candidate miRNAs and mRNAs

The selected miRNAs and mRNAs should have a strong association with both BC and other cancers in the literature (keywords such as "gene name, miRNA name, cancer, breast cancer, breast" were searched in the PubMed database),

A distinct altered expression level of these mRNAs should be detected across all of the PAM50 subgroups from normal-like to the BLBC with an increasing trend in the form of a pan flute. Genes that do not meet this criterion will be eliminated and genes with the top significant log fold change (logFC) values will be kept,

The selected genes should be associated with poor OS and DFS in BC.

## 2.4 Survival analysis

The survival data of the TCGA dataset was obtained from the Pan-Cancer Clinical publication [22]. Survival curves were obtained according to the Kaplan-Meier method (*surv_plot* function) from the *finalfit* (v1.0.3) R package [23], and differences between survival distributions were assessed by log-rank test. The patients were divided into two risk groups as high and low according to their normalized median expression values. The normalized expression values were obtained using *voom* function from *limma* (v3.46.0) R package [24]. For analysis of relationships between the selected gene and BC, univariate models were fitted using cox proportional hazard regression (*coxph* function) from the *survival* R package [25]. Furthermore, we

used the GSE25066 dataset from GEO to validate our survival analysis findings [26]. The GSE25066 dataset contains 508 BC (189 BLBC, 37 HER2-positive, 160 LumA, 78 LumB, 44 normal-like) mRNA samples.

## 2.5 The relation of circRNA, miRNA and mRNA

The circRNA–miRNA–mRNA regulatory network was established using the *Cytoscape* tool (v3.9.0) [27] based on the interactions between circRNA, miRNA, and mRNA obtained from *CSCD*, *mirDB*, *miRTarBase*, and *miRWalk* databases. After finding the DECs shared between the two circRNA datasets and DEMs shared between METABRIC and TCGA, we searched for the presence of the circRNA-miRNA pairs from the *CSCD* v2.0 database. We also checked the direction of the regulation of the DE miRNAs and DE circRNAs, since they are expected to be inversely related. This election process has shortlisted the number of DECs and their matching DEMs as described in the Results. Similarly, the presence of the matching miRNA-mRNA pairs was searched from the *mirDB*, *miRTarBase*, and *miRWalk* databases and this process has led to narrowing down the DEG list.

## 2.6 Correlation analysis between the selected miRNA and mRNAs

Spearman correlation was used to measure the correlation between the selected miRNA and mRNA expressions in the TCGA dataset. The *corrplot* function from the *corrplot* (v 0.92) [28] R package was used to visualize the correlation heatmap.

## 2.7 Analysis of the protein–protein interaction (PPI) network

The PPI network was created by using the *STRING 2021* [29] database with a minimum required interaction score of $> 0.4$ and visualized by the *Cytoscape* tool (v3.9.0).

## 2.8 Gene Ontology (GO) and Kyoto Encyclopedia of Genes and Genomes (KEGG) enrichment analyses

The gene set enrichment analyses were obtained by using *Enrichr* [30] web tool with the criterion of FDR value lower than 0.05 according to *GO* annotation and *KEGG* Pathway. *Enrichr* is a gene list enrichment analysis tool that is frequently used in the literature and allows querying on hundreds of gene sets such as *KEGG*, *GO*, *Reactome* [31], and *DisGeNet* [32]. The p-value, provided by *Enrichr*, as a result of the enrichment analysis is determined by Fisher's exact test (hypergeometric test), which is a binomial proportionality test that assumes the binomial distribution and independence for the probability of any gene set. Also, the FDR value, provided by *Enrichr*, is calculated using the Benjamini-Hochberg method to adjust the multiple hypothesis testing.

# 3 Results and discussion

## 3.1 Determination of DECs, DEMs and DEGs

**3.1.1 DECs.** We observed that 149 circRNAs (94 of them were up-regulated and 55 were down-regulated) in GSE101124 (Fig 2A) and 993 circRNA (665 of them were up-regulated and 328 were down-regulated) in GSE182471 (Fig 2B) in BC tumor samples were differentially expressed when compared to the control samples. Furthermore, we obtained 13 down- and up-regulated circRNAs in total that are shared between the GSE101124 and GSE182471 datasets. The overlapped 11 up- and two down-regulated circRNAs are listed in Fig 2C. The expression of the overlapped up- and down-regulated circRNAs in each dataset is demonstrated in S1 Fig of S1 File.

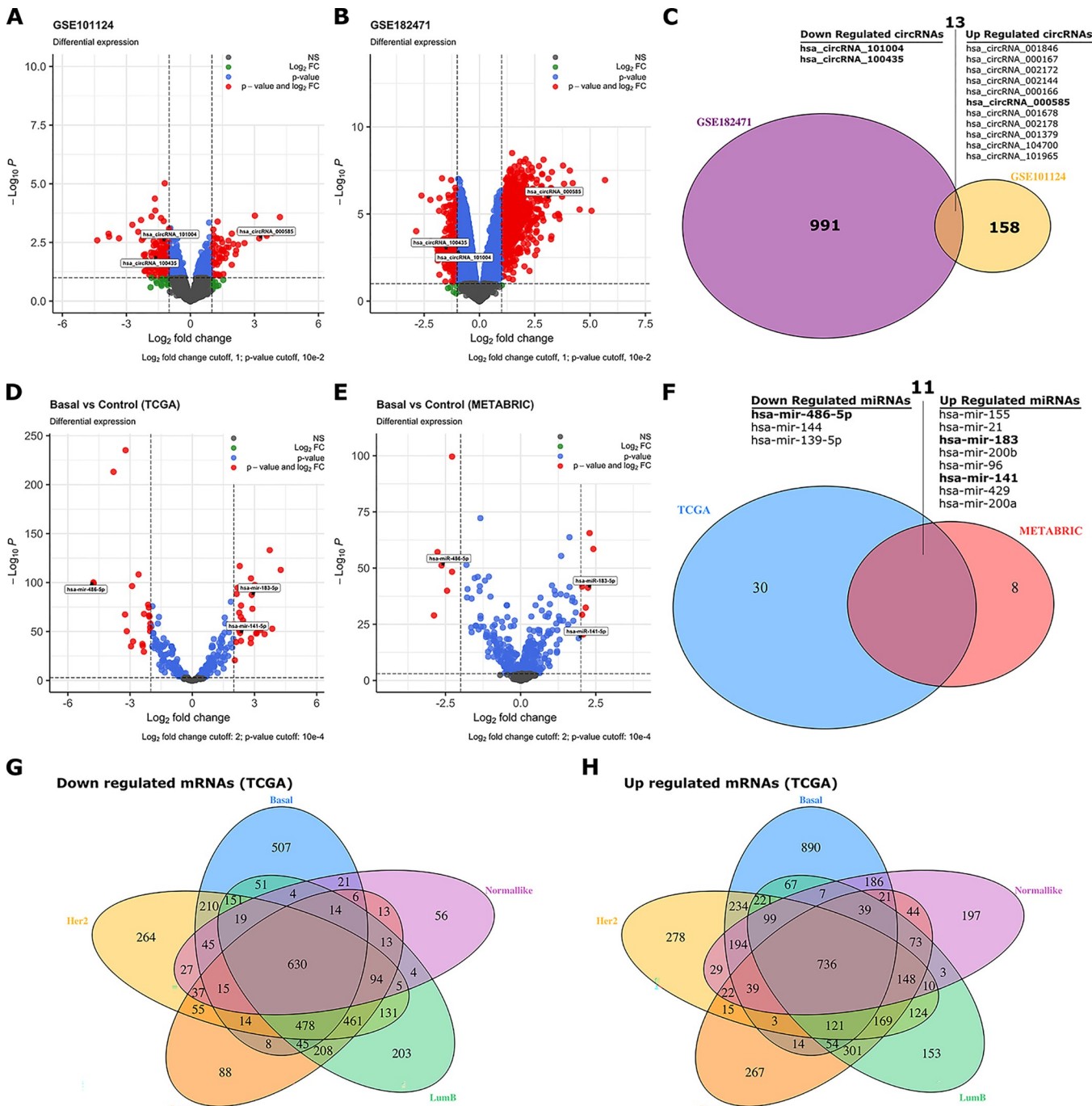

**Fig 2. The volcano plot for DECs in BC based on the two microarray datasets from GEO and intersected up- and down regulated circRNAs.** The volcano plot for DEMs in BC based on the two microarray datasets from TCGA and EGA, and intersected up- and down regulated miRNAs. (A): GSE101124, (B): GSE182471, (C): The intersected up- and down-regulated circRNAs between the GSE101124 and the GSE182471 datasets. (D): TCGA, (E): METABRIC, (F): The intersected up- and down-regulated miRNAs from shared miRNAs in the TCGA and the METABRIC datasets, The intersection of the down-regulated mRNAs (G) and the up-regulated mRNAs (H) between PAM50 subtypes, DECs: differently expressed circRNAs, BC: Breast cancer, DEMs: Differentially expressed miRNAs, EGA: European Genome-phenome Archive, BC: Breast cancer, hsa: Homo-sapiens.

**3.1.2 DEMs.** Regarding the miRNAs, in the TCGA dataset 133 miRNAs (66 of them were up-regulated and 67 were down-regulated) in BLBC samples (Fig 2D, shown as an example subtype), 114 miRNAs (65 up- and 49 down-regulated) in HER2-positive samples, 105 miR-NAs (49 up- and 56 down-regulated) in LumA group, 133 miRNAs (69 up- and 64 down-regulated) in LumB group, and 78 miRNAs (41 up- and 37 down-regulated) in normal-like tumor group were differentially expressed, when compared to the control samples.

Furthermore, in the METABRIC dataset 69 miRNAs (34 of them were up-regulated and 35 were down-regulated) in BLBC samples (Fig 2E, shown as an example subtype), 66 miRNAs (31 up- and 35 down-regulated) in HER2-positive samples, 51 miRNAs (28 up- and 23 down-regulated) in LumA group, 70 miRNAs (31 up- and 39 down-regulated) in LumB group, and 31 miRNAs (17 up- and 14 down-regulated) in normal-like tumor group were differentially expressed, when compared to the control samples.

Finally, we obtained the overlapped DEMs between the TCGA and METABRIC datasets and that are shared miRNAs across all PAM50 subtypes. The determined miRNAs (three down- and eight up-regulated miRNAs) are given in Fig 2F.

**3.1.3 DEGs.** We observed that 5,143 genes (2,925 of them were up-regulated and 2,218 were down-regulated) in BLBC samples, 5,078 genes (2,442 up- and 2,636 down-regulated) in HER2-positive samples, 4,245 genes (2,066 up- and 2,179 down-regulated) in LumA group, 4,836 genes (2,325 up- and 2,511 down-regulated) in LumB group, and 2,850 genes (1,847 up- and 1,003 down-regulated) in normal-like tumor group were differentially expressed, when compared to the control samples.

Among the down-regulated genes (Fig 2G), 630 were shared across all five PAM50 classes, whereas 736 up-regulated genes (Fig 2H) were shared across all five PAM50 classes in the TCGA dataset.

## 3.2 Determining the relationship between the detected DECs, DEMs and DEGs

Knowledge-driven investigation between the shared mRNA and miRNAs, across all five PAM50 classes, revealed that 188 up-regulated genes are associated with the 3 down-regulated miRNAs; whereas 317 down-regulated genes are found to be associated with the 8 up-regulated miRNAs, based on the *miRDB*, *miRWalk*, and *miRTarBase* databases.

## 3.3 Identification of the circRNA–miRNA interactions

The overlapped DECs were selected for further analysis. To indicate whether the 13 circRNAs (described in Section 3.1.1) play a significant role in BC, we gathered their potential target miRNAs from the *CSCD* v2.0 online databases. In the GSE101124 dataset, compared to the normal group, *hsa_circRNA_100435* and *hsa_circRNA_101004* were down-regulated in the TNBC group, while *hsa_circRNA_000585* was up-regulated (p<0.05; logFC>1.5). In total, three circRNA–miRNA interactions including three circRNAs (*hsa_circRNA_000585*, *hsa_circRNA_101004*, and *hsa_circRNA_100435*) and three miRNAs (*miR-486-5p*, *miR-141-5p*, and *miR-183-5p*) were identified in the database. *MIENTURNET* [33] was used to investigate the signaling pathways (*KEGG*, *Reactome*, *WikiPathways*, and *Disease Ontology*) in which the three miRNAs may be involved according to *miRTarBase* database. As shown in S2 Fig of S1 File, all three miRNAs were associated with some cancer-related pathways.

The basic features of the three circRNAs are displayed in Table 1. The main structural models of the three circRNAs are given in S3 Fig of S1 File. The unpaired two-samples Wilcoxon test results according to tumor and control samples of the selected three DECs are given in violin plots in Fig 3 for each circRNA dataset separately.

**Table 1. The basic features of the three DECs.**

| circRNA | Alias | circRNA type | Position (HG19) | Position (HG38) | Strand | Regulation | Gene symbol |
|---------|-------|--------------|-----------------|-----------------|--------|------------|-------------|
| hsa_circRNA_100435 | hsa_circ_0016201 | exonic | chr1:205156545\|205156934 | chr1:205187417\|205187806 | - | Down | DSTYK |
| hsa_circRNA_101004 | hsa_circ_0000375 | exonic | chr12:6657590\|6657991 | chr12:6548424\|6548825 | - | Down | IFFO1 |
| hsa_circRNA_000585 | hsa_circ_0000515 | sense overlapping | chr14:20811305\|20811534 | chr14:20343146\|20343375 | - | Up | RPPH1 |

The unpaired two-samples t-test results according to basal and control samples of the selected three (*miR-141-5p*, *miR-183-5p*, and *miR-486-5p*) DEMs are given as violin plot in Fig 4 for each miRNA dataset separately.

### 3.4 Identification of circRNA–miRNA–mRNA association

We investigated the miRNA and mRNA associations by using the *miRDB* (v6), *miRTarBase* (r8.0), *miRWalk* (v3) databases for the intersected miRNAs and shared DEGs with an absolute log2FC values greater and equal than 1. Then, we combined the circRNA–miRNA interactions and miRNA–mRNA interactions to identify the circRNA–miRNA–mRNA associations. Finally, we established a circRNA-miRNA-mRNA network, which ensures a preliminary insight into the links between the three circRNAs (*hsa_circRNA_000585*, *hsa_circRNA_101004*, and *hsa_circRNA_100435*), the three miRNAs (*miR-486-5p*, *miR-141-5p*, and *miR-183-5p*) and the 339 mRNAs. The constructed network can be seen in Fig 5.

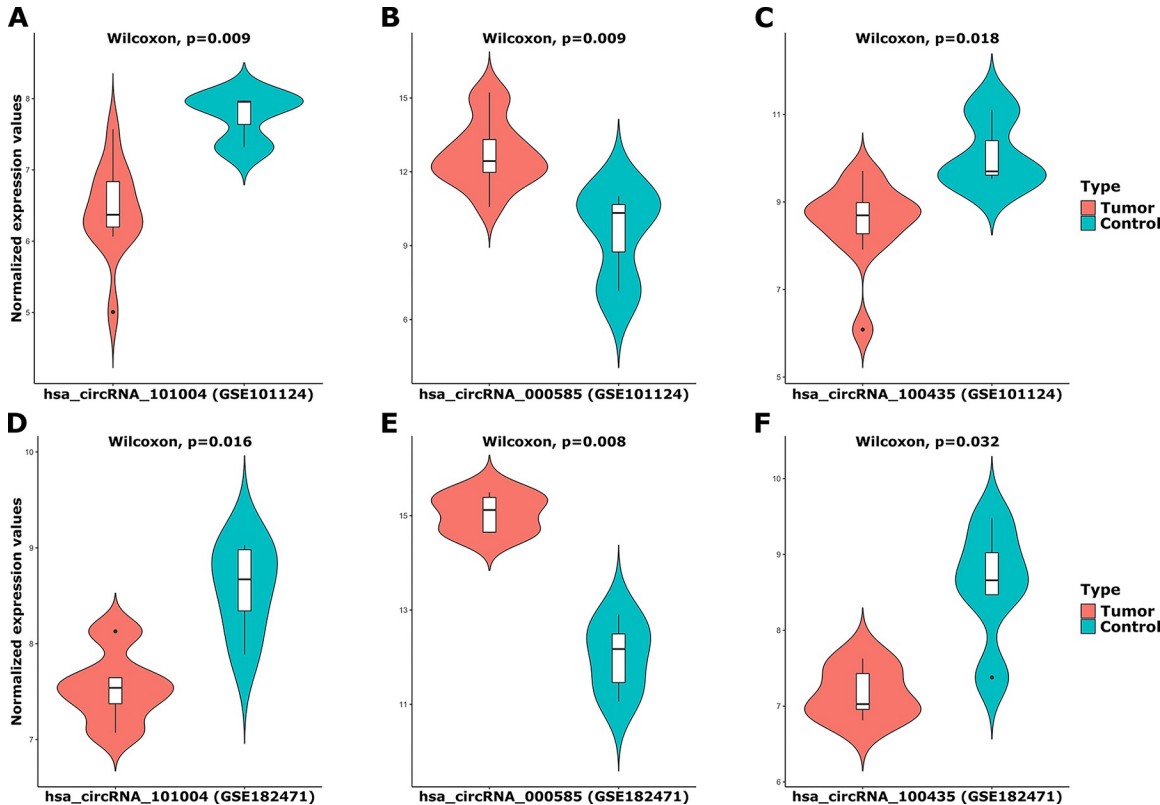

**Fig 3. The combined violin and box plots for the normalized expression values of *hsa_circRNA_101004*, *hsa_circRNA_000585*, *hsa_circRNA_100435* in GSE101124 and GSE182471 datasets by the unpaired two-samples Wilcoxon test according to tumor and control samples.** (A): *hsa_circRNA_101004* in GSE101124, (B): *hsa_circRNA_000585* in GSE101124, (C): *hsa_circRNA_100435* in GSE101124, (D): *hsa_circRNA_101004* in GSE182471, (E): *hsa_circRNA_000585* in GSE182471, (F): *hsa_circRNA_100435* in GSE182471.

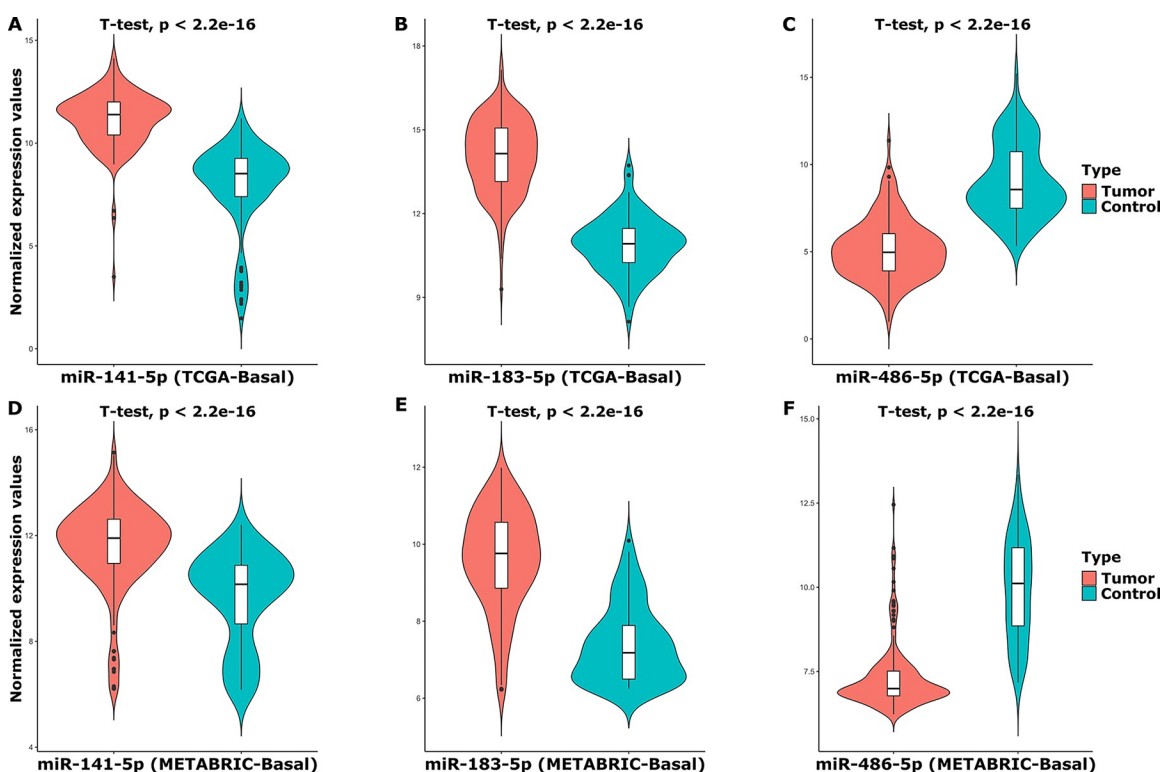

**Fig 4. The combined violin and box plots for the normalized expression values of *miR-141-5p*, *miR-183-5p*, and *miR-486-5p* in TCGA and METABRIC datasets by the unpaired two-samples t-test according to basal and control samples.** (A): *miR-141-5p* in TCGA, (B): *miR-183-5p* in TCGA, (C): *miR-486-5p* in TCGA, (D): *miR-141-5p* in METABRIC, (E): *miR-183-5p* in METABRIC, (F): *miR-486-5p* in METABRIC.

Among the DE 339 mRNAs, we selected the top 10 genes for each one of the three selected miRNAs individually according to the log2FC values and that are the target of the shortlisted three miRNAs. After that, we only kept the mRNAs which demonstrate an increasing logFC trend in the form of a pan flute from normal-like to the BLBC (Fig 6). Hence, according to the criteria we defined (criterion I and II), a distinct altered expression level of 18 genes (*SDC1*, *PRAME*, *MELK*, *NEK2*, *EXO1*, *TPX2*, *BUB1*, *DLGAP5*, *CIDEC*, *ADH1B*, *TMEM132C*, *ACVR1C*, *LIPE*, *ABCA8*, *BTNL9*, *TNXB*, *GPAM*, and *AOC3*) were detected in PAM50 sub-groups from the normal-like group to the BLBC group (expression levels shown in S4 Fig of S1 File). We demonstrated the Log2FC values of these 18 DEGs in S2 Table and in the form of barcharts in Fig 6. This splits the genes into three groups according to their target miRNAs. BLBC has the greatest Log2FC values for 13 out of 18 DEGs (almost all target genes of *miR-486-5p* and *miR-183-5p*) and the bar charts represents almost like a pan flute form, with the Log2FC values represent a decreasing trend from BLBC towards HER2, LumB, LumA, and Normal-like subtype. We also showed the correlation patterns between shortlisted three DEMs and their target shortlisted 18 DEGs (Fig 7 represents correlation patterns (A) across all samples and (B) across solely BLBC-subtype samples).

## 3.5 Survival analysis

We performed survival analysis on the shortlisted 18 genes (S5 and S6 Figs in S1 File for overall and disease-free survival, respectively, in the TCGA dataset). We observed that only four genes (*SDC1*, *DLGAP5*, *PRAME*, and *EXO1*) demonstrated a significant survival outcome in both

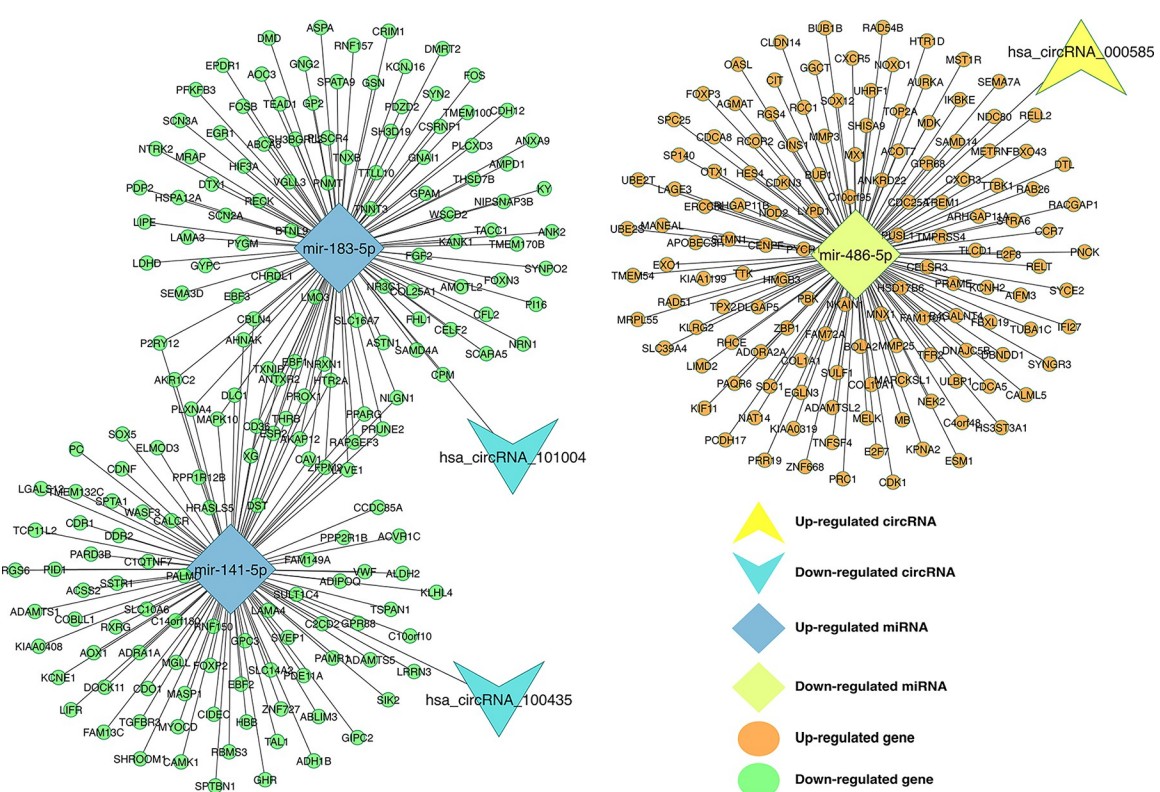

**Fig 5. circRNA–miRNA–mRNA regulatory network.** The network consisting of three cricRNAs (*hsa_circRNA_000585*, *hsa_circRNA_101004*, and *hsa_circRNA_100435*), three miRNAs (*miR-486-5p*, *miR-141-5p*, and *miR-183-5p*) and 339 genes was generated by Cytoscape 3.9.0.

TCGA and GSE25066 datasets. The increased expression of the *SDC1* gene had a poor disease-free survival (DFS) and overall survival (OS) and over-expressions of *DLGAP5*, *PRAME*, and *EXO1* genes had a poor DFS (Fig 7C–7E and S7A-S7C Fig in S1 File) in the TCGA dataset. We also found that highly expressed *SDC1* had a poor DFS, and over-expressions of *DLGAP5*, *PRAME*, and *EXO1* genes had a poor DFS (S7D-S7F Fig in S1 File) in the GSE25066 [26] dataset. The expression distributions of the *SDC1*, *PRAME*, *EXO1*, and *DLGAP5* genes are shown in S8 Fig of S1 File.

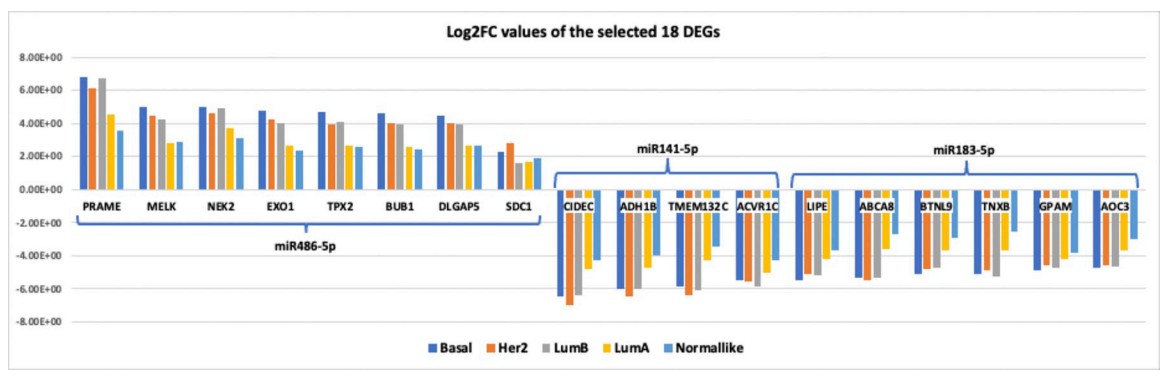

**Fig 6. Log2FC values of the selected 18 DEGs.**

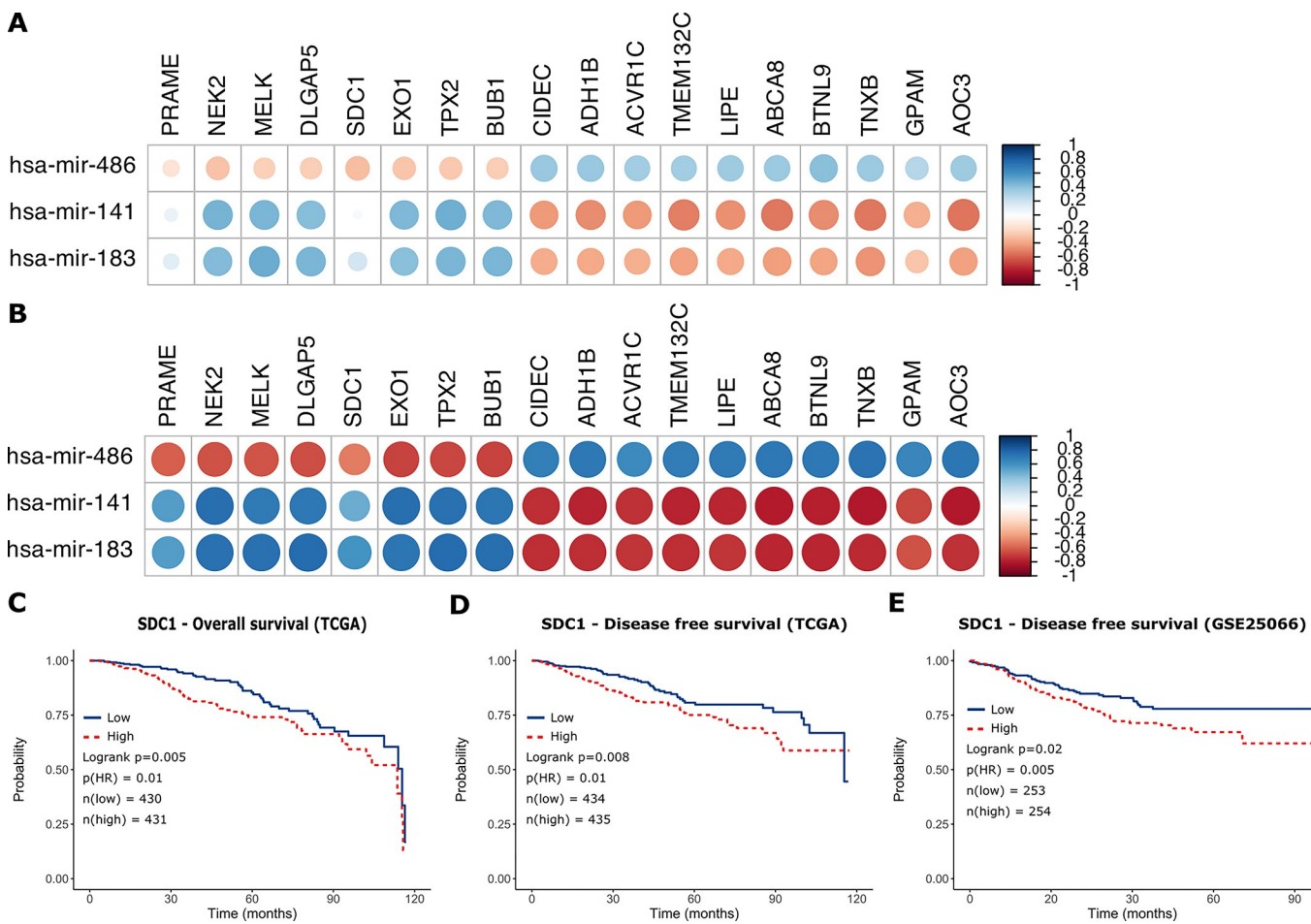

**Fig 7.** The correlation heatmap of the selected mRNA and miRNAs (A):All PAM50 groups, (B): Only BLBC subgroup, Survival analysis of *SDC1*. (C): Overall survival of *SDC1* in TCGA dataset, (D): Disease-free survival of *SDC1* in TCGA dataset, (E): Disease-free survival of *SDC1* in GSE25066 dataset.

### 3.6 Identification of the hub genes with bottleneck algorithm from the PPI network

Using the genes in Fig 5, after removing the isolated nodes, we constructed a PPI network, which consists of 138 nodes and 780 edges, to observe the interactions among the 339 mRNAs (Fig 8A). Considering the importance of the hub gene in a network, we utilized the bottleneck algorithm to screen hub-genes from the PPI network. The subnetwork with 14 nodes (10 hub genes and 4 extended genes) and 24 (14 between hub genes and 10 between extended genes) edges was determined (Fig 8B), which uncover the crucial roles of the ten genes (*AHNAK, CAV1, CDK1, EGR1, FGF2, FOS, KIF11, PPARG, SDC1,* and *TNXB*) in BC. A circRNA-miRNA-hub gene network was then built to describe the links among the DECs, DEMs and hub genes (Fig 9). Thirteen circRNA–miRNA–mRNA regulatory modules, including *hsa_-circRNA_100435/ miR-141-5p/ AHNAK* regulatory axis, *hsa_circRNA_100435/ miR-141-5p/ PPARG* regulatory axis, *hsa_circRNA_100435/ miR-141-5p/ CAV1* regulatory axis, *hsa_-circRNA_101004/ miR-183-5p/ AHNAK* regulatory axis, *hsa_circRNA_101004/ miR-183-5p/ PPARG* regulatory axis, *hsa_circRNA_101004/ miR-183-5p/ CAV1* regulatory axis, *hsa_-circRNA_101004/ miR-183-5p/ FGF2* regulatory axis, *hsa_circRNA_101004/ miR-183-5p/*

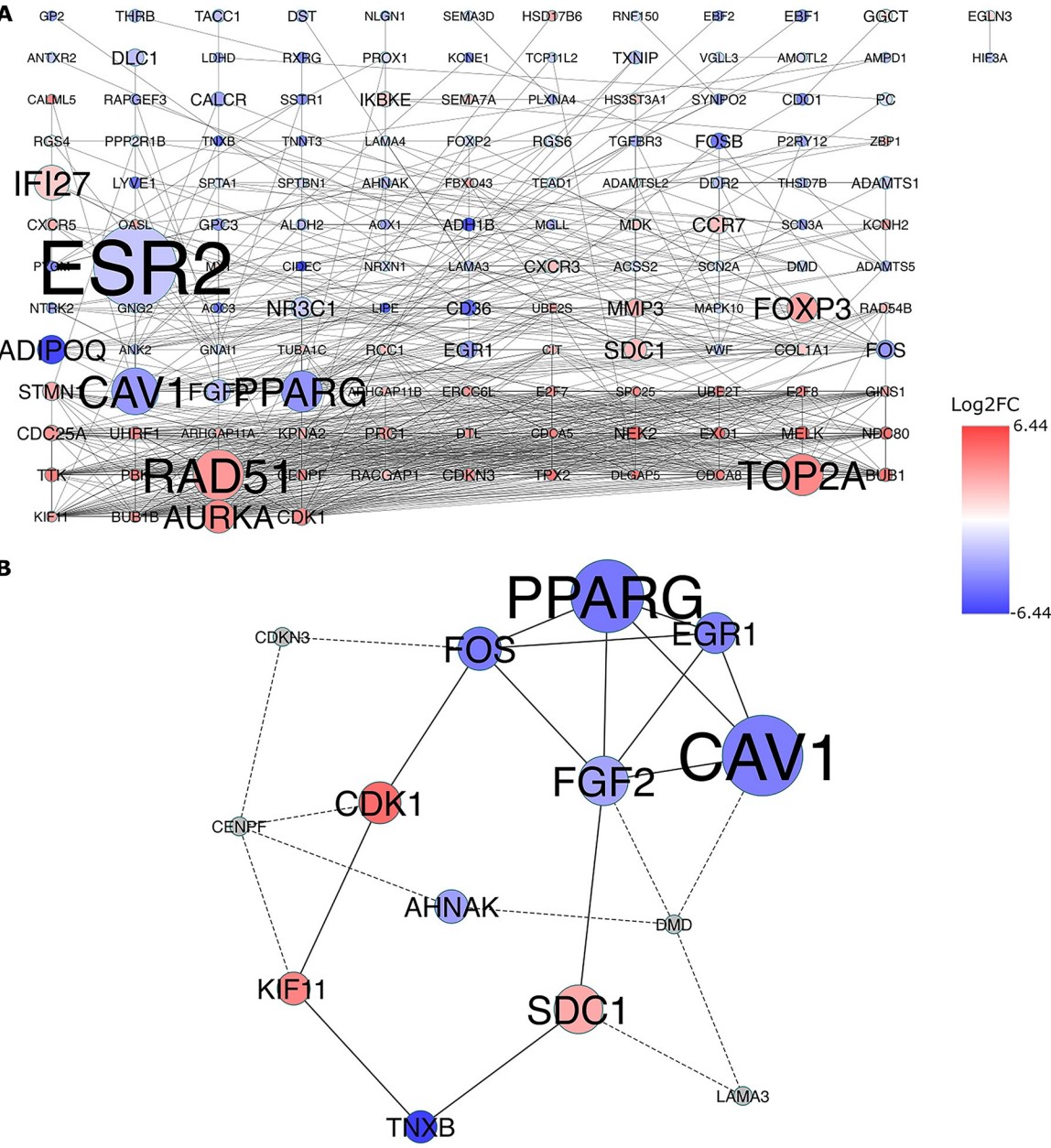

**Fig 8. Identification of hub genes from the PPI network by bottleneck algorithm using the cytoHubba Cytoscape plugin.** The node color changes gradually from blue to red in ascending order according to the log2 (fold change) of genes. (A): A PPI network of the 339 target genes playing crucial roles in BC. This network consists of 138 nodes and 780 edges. The node size changes gradually from small to large in ascending order according to the number of the PMIDs from DisGeNET per gene. (B): A PPI network consist of the ten hub genes (colored blue and red) and 4 extended genes (colored gray) extracted from a. This network consists of 14 (10 hub genes and 4 extended genes) nodes and 24 (14 between hub genes and 10 between extended genes) edges. PPI protein–protein interaction, BC: Breast Cancer.

*EGR1* regulatory axis, *hsa_circRNA_101004/ miR-183-5p/ TNXB* regulatory axis, *hsa_-circRNA_101004/ miR-183-5p/ FOS* regulatory axis, *hsa_circRNA_000585/ miR-486-5p/ CDK1* regulatory axis, *hsa_circRNA_000585/ miR-486-5p/ KIF11* regulatory axis, and *hsa_-circRNA_000585/ miR-486-5p/ SDC1* regulatory axis, were found from the network.

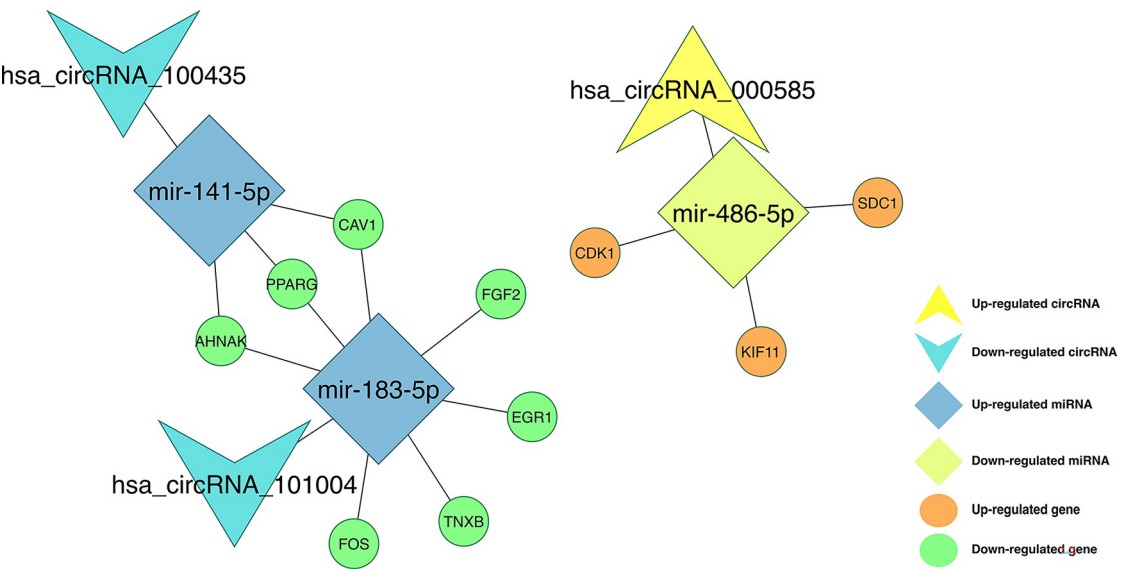

**Fig 9. CircRNA–miRNA–hubgene network.** The network consisting of three circRNAs (*hsa_circRNA_000585*, *hsa_circRNA_101004*, and *hsa_circRNA_100435*), three miRNAs (*miR-486-5p*, *miR-141-5p*, and *miR-183-5p*) and 10 hub genes (*AHNAK, CAV1, CDK1, EGR1, FGF2, FOS, KIF11, PPARG, SDC1*, and *TNXB*) was generated by Cytoscape 3.9.0.

### 3.7 GO annotation and KEGG pathway analyses of the ten hub genes

GO analysis was performed to demonstrate the functional annotations of the ten hub genes. The top five highly enriched GO terms related to biological process (BP), cellular component (CC) and molecular function (MF) are shown in Fig 10A. The most enriched GO terms in BP was "positive regulation of pri-miRNA transcription by RNA polymerase II (GO:1902895)" (FDR = 7.56E-07), that in CC was "sarcolemma (GO:0042383)" (FDR = 8.51E-03), and that in MF was "transcription regulatory region nucleic acid binding (GO:0001067)" (FDR = 7.74E-03). KEGG pathway analysis was carried out to determine the signaling cascade in which ten genes are involved. With an FDR < 0.05, 17 significantly enriched pathways were determined

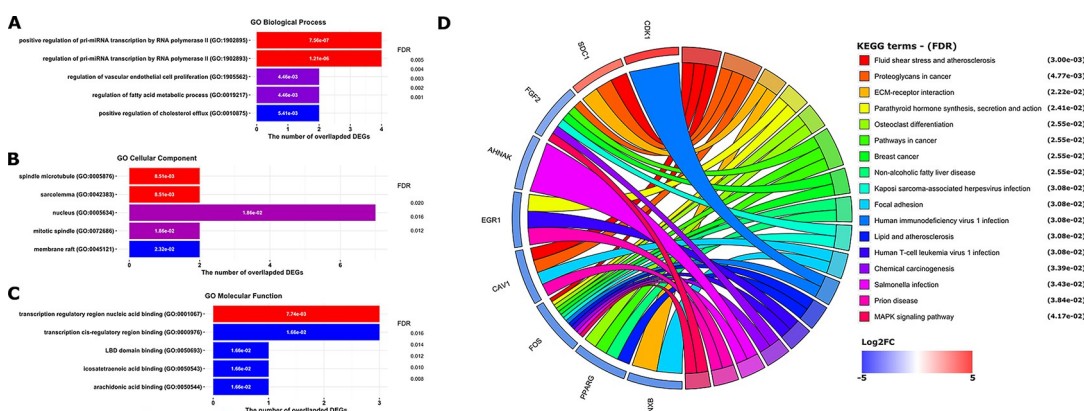

**Fig 10.** Top five Gene Ontology (GO) enrichment annotations of the ten hub genes: (A): biological process, (B): cellular component, (C): molecular function. (D)The significantly enriched Kyoto Encyclopedia of hub-genes and genomes (KEGG) pathways with a FDR < 0.05. The results of the GO and the KEGG analyses were obtained from the 'Enrichr' web tool (https://maayanlab.cloud/Enrichr/) and visualized by R package 'ggplot2'. Cohort plot shows that the ten hub genes are correlated via ribbons with their assigned KEGG terms. FDR: False discovery rate, is calculated using the Benjamini-Hochberg method to adjust the multiple hypothesis testing.

(Fig 10B). Among the 17 pathways, "Proteoglycans in cancer pathway" and "Breast cancer pathway" are linked with the BC progression [34, 35]. In addition, some of the pathways such as "Pathways in cancer", "Chemical carcinogenesis", and "Non-alcoholic fatty liver disease" were also tumor-related pathways.

## 3.8 Summary of the filtered circRNA/miRNA/gene axis

The DE circRNAs with Log2FC value ≥1 were intersected in both datasets, GSE101124 and GSE182471, and three circRNA remained (*hsa_circ_0000515*, *hsa_circ_0016201*, and *hsa_circ_0000375*) according to their existence in the *CSCD* database. Similarly, overlapping DE miRNAs with Log2FC value ≥2 in both TCGA and METABRIC, were kept. Among them, three miRNAs (*miR-486-5p*, *miR-141-5p*, and *mir-183-5p*) were presented in the *CSCD* database as the targets of the three shortlisted circRNAs.

Then, the 18 DE genes with a Log2FC value of ≥2 which were reported to be strongly associated with BC, and could be targeted by the shortlisted miRNAs, were determined. Those genes were found to be differentially expressed for all PAM50 groups but most significantly in the BLBC subtype (as shown in Fig 6 the logFC values are highest for the BLBC group). Additionally, as shown in Fig 7, correlations between the mRNAs and the target miRNAs are stronger in the BLBC-subtype samples (Fig 7B) compared to the correlation patterns obtained by using all tumor samples together (Fig 7A).

The possible circRNA-miRNA-mRNA interaction, which was detected to play a role in the cellular processes of BC, is shown in Fig 11.

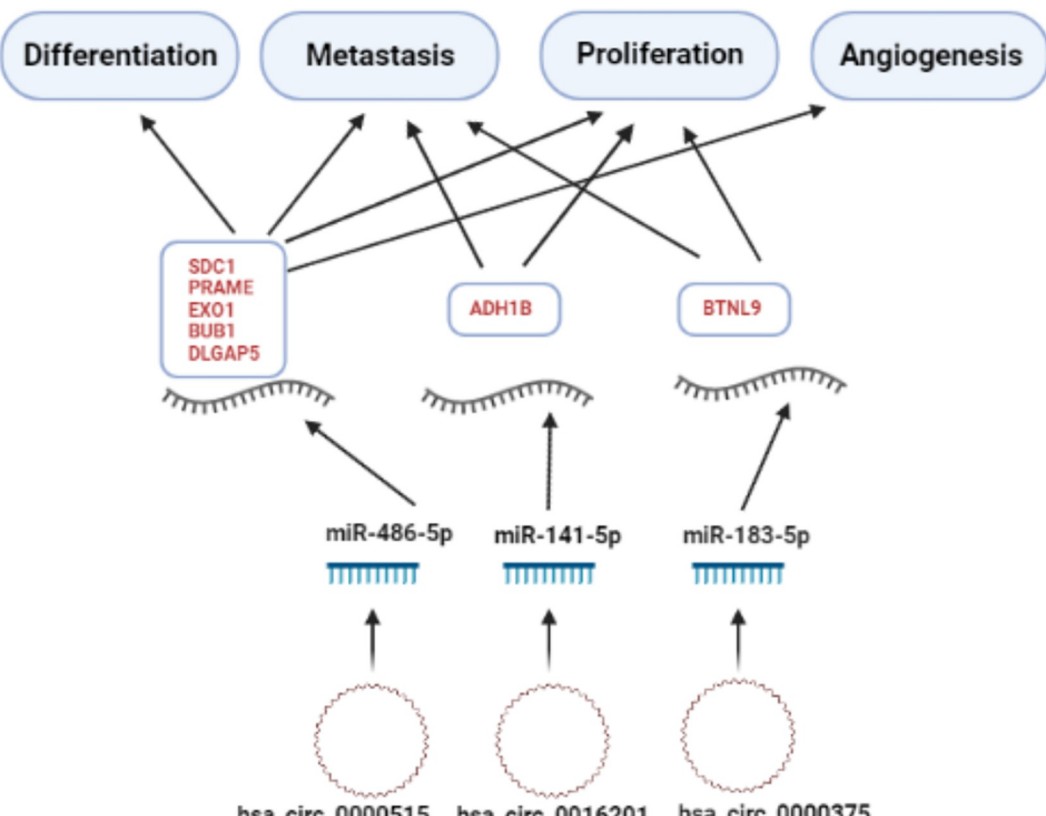

**Fig 11. The summary of the possible role of circRNA/miRNA/gene axis in BC pathogenesis according to our study.**

## 3.9 Discussion

BC is the most frequently identified tumor among women around the world and more than 90% of BC deaths are related to metastasis. Existing treatment approaches for metastatic BC have been inadequate, compounded by a lack of early prognosis/ predictive criteria for estimating which body parts are most susceptible to metastasis. Although there are many new developments in the fields such as chemotherapy, endocrine treatment and targeted therapy for BC in recent years, this cancer type is still the most common cancer in women with high morbidity and mortality [36]. Subtypes in BC are heterogeneous and treatment practices are determined according to these subtypes. From better to the worst, the aggressiveness of the BC subgroups are generally in the following order: Normal breast-like, LumA, LumB, HER2-positive, BLBC [37, 38]. It is clearly known that the OS of cases with HER2-positive and BLBC groups are worst in PAM50 subtypes [39]. The cells are "basal-like," which implies they match the basal cells which line the breast ducts. It is strongly associated with TNBC appearance described by the deficiency of expression of ER, PR, and HER2-positive. BLBC, which is more associated with distant metastasis, has an extremely poor prognosis compared to other intrinsic BC groups, and the success in its treatment is currently limited [40, 41]. This knowledge has substantially advanced our understanding of BC's heterogeneity and the several biological processes that the disease employs. In 2009, Parker et al. defined a minimum gene set, PAM50, for categorizing these intrinsic subgroups [42]. Because the biology of all these intrinsic subgroups indicates changes in incidence, responsiveness to therapy and survival, unique genes for each subtype may be evaluated as markers to direct potential treatments. In this respect, it is crucial to elucidate novel circRNA-miRNA-mRNA relationships in the determination of these subgroups [43–45]. In our study, the expression states of miRNAs and genes in datasets were classified according to the five molecular subtypes classification.

circRNAs, which are a new class of endogenous evolutionarily conserved RNAs, have a stable structure and they are stated to serve as vital regulators in the various cellular activities. According to studies conducted so far, it has been understood that the main reason for circRNAs to act as critical regulators in cells is their relationship with target miRNAs [46]. In recent years, it has been suggested that miRNAs act as a bridge in the realization of the role of circRNAs in the regularization of cellular events [47]. circRNAs change gene expression by acting as miRNA sponges with their binding sites [3, 48]. As increased expression rates of circRNAs in the cell may contribute to decreased expressions of target miRNAs and the increased expressions of target genes. circRNA-miRNA-mRNA interactions, which are the focus of this work, are very new to the scientific world but experiments have shown that these relationships could be beneficial for the detection of novel biomarkers for cancer [49, 50]. The studies on circRNAs about the determination of subtypes of BC are limited. The study by Nair et al. in 2016 is one of the first studies showing circRNAs may be useful in identifying subtypes of BC [51]. In the study of Darbeheshti et al. in 40 TNBC, 20 Lum A, 18 Lum B and 17 HER2-positive tumor samples, it was determined that *hsa_circ_0044234* has a distinct molecular signature as a potential *GATA3* regulator in TNBC [52]. In another study, *circ-PGAP3* was shown to increase TNBC proliferation and invasion via *miR-330-3p/Myc* axis [53]. Sheng et al have also found out overall BC-associated circRNA-miRNA-mRNA interactions but they have not investigated those interactions for PAM50 [18] subtypes which is different from our study design.

As a result of our study, many circRNAs, miRNAs and genes that may be associated with BC have been identified. We found that three circRNAs (*hsa_circ_0016201*, *hsa_circ_0000375*, and *hsa_circ_0000515*), three miRNAs (*mir-183-5p*, *miR-141-5p*, and *miR-486-5p*) and 18 genes (*CIDEC*, *ADH1B*, *TMEM132C*, *ACVR1C*, *LIPE*, *ABCA8*, *BTNL9*, *TNXOCB3*, *GPAM*,

*PRAME*, *MELK*, *NEK2*, *EXO1*, *TPX2*, *BUB1*, *DLGAP5*, and *SDC1*) may be important in BC, especially in a basal-like group, by applying filters as described in the material method section. Analysis of TNBC versus normal tissue samples on the GSE101124 dataset results may indicate that these three circRNAs highlighted in the study may play an important role in BLBC. Moreover, although the expression alteration of genes, targeted by miRNAs that sponged via selected circRNAs, was more prominent in the BLBC and Her2-positive subgroups, it was much more limited in the normal-like and Lum-A groups (an expression alteration was detected as similar to a pan flute, Fig 6).

Expressions of *miR-141-5p* and *mir-183-5p*, which are known to be dysregulated in many cancers including BC (46–50), were found to be significantly increased in our study in all dataset samples from all PAM50 groups. Possible target genes that may contribute to the cancer progression and in which *miR-141-5p* and *mir-183-5p* could alter their expression in BC are shown in S1 Table. According to the criteria we determined, the possible paired targets of *miR-141-5p*/ *ADH1B* and *mir-183-5p*/ *BTNL9*, may be related to the BC process. In the dataset we examined, it was identified that *hsa_circ_0016201*, which is among the circRNAs whose expression was significantly decreased, could have a role as a sponge for *miR-141-5p* and *hsa_-circ_0000375* could be acted as a sponge for *miR-183-5p*. Therefore, we would like to emphasize that the relationship between *hsa_circ0016201*/ *miR-141-5p*/ *ADH1B* and *hsa_0000375*/ *miR-183-5p*/ *BTLN9* should be investigated at the cellular functional level. It could be substantial to examine this relationship with conventional molecular genetic techniques in both BC cells and tumor tissue.

More importantly, the expression of *miR-486-5p*, which is an essential tumor suppressor miRNA in BC and many other cancer types [54–57], was significantly decreased in all PAM50 groups examined in our study. It was determined that *hsa_circ_0000515*, one of the circRNAs whose expression was significantly increased in the dataset we detected, could act as a sponge for *miR-486-5p*. In addition, we determined that the increased expression of *SDC1*, *PRAME*, *EXO1*, *BUB1*, and *DLGAP5* genes could be more strongly associated targets of *miR-486-5p* in BC. The overexpressed *SDC1* gene was found to lead a significantly poor OS and DFS and overexpressed *PRAME*, *EXO1*, *BUB1* and *DLGAP5* genes were found to lead a significantly poor DFS in BC (Fig 7C–7E and S8 Fig in S1 File) (criterion III). *miR-486-5p* has been notified as an important tumor suppressor miRNA in various cancers, including BC. It has been reported that *miR-486-5p* which could be found exosomal miRNA in BC inhibits epithelial-mesenchymal transition (EMT) by targeting Dock1 and suppresses cancer cell proliferation by targeting the *PIM-1* oncogene in BC, can be used as a biomarker in the prediction of BC recurrence [56–58]. Valuable studies are showing that *miR-486-5p* may be associated with different circRNAs. The importance of *circNFIB1*/*miR-486-5p*/*PIK3R1*/*VEGF-C* axis in lymphatic system metastasis in pancreatic cancer [59]; *hsa_circ_0016788*/ *miR-486-5p*/ *CDK4* pathway in hepatocellular carcinoma tumorigenesis [60]; *circHUWE1*/*miR-486-5p* in colorectal cancer migration and invasion [61] and *Circ-TCF4.85*/ *miR-486-5p*/ *ABCF2* in hepatocellular carcinoma progression [62] are reported in the literature. However, as far as we know the relationship between *miR-486-5p* and circRNAs has not yet been reported in BC. Syndecan-1 (*SDC1*, *CD138*) is a critical cell surface adhesion molecule required for cell morphology and impact on the natural microenvironment. *SDC1* dysregulation enhances cancer development by increasing cell proliferation, angiogenesis, invasion, and metastasis and is linked to chemo-resistance. *SDC1* expression is also correlated to chemotherapy responses and prognosis in numerous solid and/ or hematological cancers, including BC [63, 64]. It has been suggested that *SDC1* could be a new molecular marker that alters the phenotype of cancer stem cells through the *IL-6*/*STAT3*, Notch, and EGFR signaling pathways in triple-negative inflammatory BC [65]. Induction of *SDC1* in the lung microenvironment may promote the formation of breast tumor

metastasis [66]. *SDC1* has been found to have a prominent role in the process of BC metastasis to the brain. *SDC1* has been shown to increase BC cell migration across the blood-brain barrier via modulating cytokines, which may alter the blood-brain barrier [67]. *SDC1* overexpression in BC is associated with various miRNAs [68, 69]. However, the relationship between *miR-486-5p/ SDC1* and BC is not yet known. It has been reported that *SDC1* expression can also be indirectly altered by circRNAs as it has been demonstrated that *circCEP128* is associated with bladder cancer progression via the *miR-515-5p/ SDC1* axis [70]. According to our bioinformatics study findings, we recommend further investigation of the *SDC1* gene together with the *hsa_circ_0000515/ miR-486-5p* axis when conducting circRNA/miRNA/gene functional research in BC. The *SDC1* hub gene, which is targeted by miR-486-5p, was found to be highly compatible with the selection criteria we applied in our study. We propose that *hsa_-circ_0000515/ miR-486-5p/ SDC1* axis may be an important biomarker candidate in distinguishing patients especially in the BLBC group, according to the PAM50 classification of BC.

## 4 Conclusion

Finding new biomarkers to clearly classify subtypes of BC could be quite crucial in the battle against cancer. To identify novel biomarkers and new therapeutics, a deeper understanding of the mechanisms underlying BC metastasis is extremely important. According to our study results, we suggest various DE mRNAs, miRNAs and circRNAs that may be important in the onco-transcriptomic cascade for BC. The interrelationships of these molecules can be potential diagnostic biomarkers or therapeutic targets. Therefore, functional experiments such as proliferation, apoptosis, invasion, and metastasis on BC cells should be studied to elucidate these circRNA-miRNA-mRNA relationships in the future.

## Supporting information

**S1 File. Contains all the supporting figures.**
(DOCX)

**S1 Table. Details of databases, tools, and R packages used in this study.**
(XLSX)

**S2 Table. Log-2 transformed fold change (Log2FC) values of the 18 DEGs.**
(XLSX)

## Acknowledgments

Additional information
    More detailed information about the manuscript can be found on the GitHub repository.
    (https://github.com/cihaterdogan/Breast_Cancer_RN).

## Author Contributions

**Conceptualization:** Zeyneb Kurt.

**Data curation:** Cihat Erdogan, Ilknur Suer, Murat Kaya.

**Formal analysis:** Cihat Erdogan, Ilknur Suer, Murat Kaya.

**Investigation:** Cihat Erdogan, Ilknur Suer, Murat Kaya.

**Methodology:** Zeyneb Kurt.

**Project administration:** Zeyneb Kurt.

**Supervision:** Sukru Ozturk, Nizamettin Aydin.

**Writing – original draft:** Cihat Erdogan, Ilknur Suer, Murat Kaya.

**Writing – review & editing:** Sukru Ozturk, Nizamettin Aydin, Zeyneb Kurt.

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
