## [Decision Letter · Decision Letter 0]

25 Jul 2023

PONE-D-23-03451Bioinformatics Analysis of the Potentially Functional circRNA-miRNA-mRNA Network in Breast CancerPLOS ONE

Dear Dr. Kurt,

Thank you for submitting your manuscript to PLOS ONE. After careful consideration, we feel that it has merit but does not fully meet PLOS ONE’s publication criteria as it currently stands. Therefore, we invite you to submit a revised version of the manuscript that addresses the points raised during the review process.

We look forward to receiving your revised manuscript.

Kind regards,

Demet Cansaran Duman

Academic Editor

PLOS ONE

Journal Requirements:

2. Please note that PLOS ONE has specific guidelines on code sharing for submissions in which author-generated code underpins the findings in the manuscript. In these cases, all author-generated code must be made available without restrictions upon publication of the work. 

Please review our guidelines at https://journals.plos.org/plosone/s/materials-and-software-sharing#loc-sharing-code and ensure that your code is shared in a way that follows best practice and facilitates reproducibility and reuse.

Reviewers' comments:

Reviewer's Responses to Questions

**Comments to the Author**

1. Is the manuscript technically sound, and do the data support the conclusions?

Reviewer #1: Partly

Reviewer #2: Yes

2. Has the statistical analysis been performed appropriately and rigorously? 

Reviewer #1: Yes

Reviewer #2: Yes

3. Have the authors made all data underlying the findings in their manuscript fully available?

Reviewer #1: Yes

Reviewer #2: Yes

4. Is the manuscript presented in an intelligible fashion and written in standard English?

Reviewer #1: Yes

Reviewer #2: Yes

5. Review Comments to the Author

Reviewer #1: Circular RNAs (circRNA) are involved in regulating proliferation, invasion, and migration of cancer and may serve as novel molecular markers for diagnosis, prognosis evaluation and treatment of cancer. The current study represents a bioinformatic-based analysis to identify potential circRNA-based prognostic and targeted therapy strategies for Breast Cancer (BC). Here, the authors indicated that hsa_circ_0000515/ miR-486-5p/ SDC1 axis may be an important potential biomarker for BC patients in the basal-like breast cancer (BLBC) group. The overall level of the paper is good and important issues are emphasized. However, some issues listed below need to be addressed by the authors to enhance their manuscript.

1. Authors are requested to care using full expression of some terms followed by its abbreviation in the paper.

2. To identify differentially expressed circRNAs (DECs), Authors downloaded two datasets containing circRNA expression profiles in BC patients from Gene Expression Omnibus (GEO) database. Authors are requested to give related links or references for databases or other web-based tools (for example: CSCD v2.0) that are used in the bioinformatic analysis in the materials and method section.

3. The cutoff parameters should be given for STRING analysis in the “Analysis of the protein–protein interaction (PPI) network” title of the materials and methods section.

4. The authors found differentially expressed circRNAs (DECs) by using two independent GEO datasets (GSE101124 and GSE182471). This analysis was conducted in Breast Cancer Compared to Adjacent Tissues. However, Identification of Differentially Expressed mRNAs and miRNA were performed in PAM50 breast cancer subtypes using TCGA and metabric datasets. Authors shared the mRNA and miRNA analysis for BLCL subtype. Each of the five breast cancer molecular subtypes vary by their biological properties and prognoses. The circRNAs that were found in the current study might have a different profile when we compared the expression profile as subtype specific. The Authors are asked why they didn’t compare TCGA and metabric datasets as tumor vs normal and whether this article belongs to breast cancer in general or to the BLCL subtype. Authors are requested to explain this issue.

5.There is a similar study in the literature which was published in 2021 (https://doi.org/10.1155/2021/1732176). This study could affect the novelty of this manuscript. The authors are requested to discuss this study in terms of its advantages and disadvantages in the discussion section.

6. Calculation of Spearman’s rank correlation coefficient between DE miRNA and DE circRNA/mRNA expression profiles could be better.

7. According to the ceRNA hypothesis, circRNA can act as effective ceRNAs depending on their abundance and cytoplasmic localization. Although circRNAs are enriched mainly in the cytoplasm, some studies on circRNAs indicate different subcellular localization. Therefore, subcellular localization analysis might be important for the hsa_circ_0000515/miR-486-5p/SDC1 axis. To support this axis, the authors can at least use a prediction tool such as Circ-LocNet for subcellular localization and coding potential analysis.

8. This study needs validation such as qRT-PCR for the potential candidates.

9. Similar to DEM and DEC, DEG results should also be represented in the fig 2.

10. Authors expressed the 16 DECs as the most significantly changed ciRCRNAs in the “2.2 Predicting the associated biological features” part of the results section. 16 circRNAs, overlapping circRNAs from 2 GEO circRNA datasets or the most significantly changing circRNA?

11. The selection strategy of the 3 candidate circRNA (has_circRNA_0000375, has_circRNA_0000515, has_circRNA_0016201) is not clear. Authors are requested to explain how to find these 3 circRNAs? Selection of 3 miRNAs is also unclear. Selection of the miRNA and mRNAs could be explained in the results according to section 2.2 in the materials and methods in detail. Authors indicated that 11 common miRNAs from TCGA and metabric miRNA data sets analysis. Selection of 3 miRNAs, 3 circRNAs and genes could be shared in a better way.

12. Metabric dataset also has mRNA expression analysis data. Authors could have also used this dataset for DEG analysis or for validation.

13. Authors have found 18 genes from TCGA Breast cancer mRNA data analysis according to their criteria. However, they showed the survival analysis results for four of them. It is not clear why only these four genes were represented.

Reviewer #2: In this manuscript the authors aimed to find out regulatory networks of circRNAs, miRNAs and mRNAs in breast cancer. The manuscript has an importance since finding out circuits and networks is important to understand the development and progression of breast cancer from systems biology approach. The authors have to be more precise about the hypothesis they suggested; it can not be extracted from the manuscript whether they are dealing with the networks related to the formation (T/N) or the molecular classification (PAM50) of breast cancer. And the authors should consider the points raised below to be accepted;

The authors need to revise the word “database” to data sets” in the following sentence, which is in the introduction section of the manuscript; “Therefore in our study circRNAs that may be relevant to BC were detected using the GSE101124 and GSE182471 databases.” These series are the datasets submitted to GEO database.

Although the main aim of the study is defined as to find out the regulative circRNA-miRNA-mRNA

network in breast cancer, they do not mention anything about these networks or the structure of the network or the significance of finding out networks in the introduction. Introduction is too brief and not so informative at all.

The authors have to give the details of the selection criteria of the expression data of circRNAs. Why did they use only two array data instead of adding NGS data to increase the sample number. They may explain this as a limitation of the study.

One of the data set GSE101124 is comprised of both breast cancer cell lines and breast tumors with different subtypes (Luminal A and TNBCs). And the subtypes of the tumor samples are not mentioned in the study of GSE182471. The authors need to explain the characteristics of the data they used in the manuscript rather than to give it up to the readers. As far as it is understood the authors analyzed the cell line data together with the tumor data which is completely irrelevant. They have to exclude the cell line data since their genetic makeup is completely different from the tumor samples as they are cultured, and many passages are performed.

As it can be extracted from the figure 2 but not from the text, the authors found out the differentially expressed circRNAs among all groups of tumors compared to control samples (tumors (LumA, TNBC, cell lines) vs controls) but although they found out DE miRNAs between all subtypes they only concentrated on the basal tumors and controls. Then they related these DE miRNAs and circRNAs, which are obtained from different sample types. The authors must explain the selection criteria of the samples to be compared clearly. Otherwise, the relationship they are trying to establish between circRNAs and miRNAs could not be robust.

The selection criteria of 3 DE circRNAs among 16 common DE circRNAs for further studies is not justified. The authors have to be more specific about the biological relevance of why they selected these three. And the same comment is valid for the selection of DE 3 miRNAs. They need to explain it in the results section more clearly.

As far as I understand the authors selected DE genes by comparing all subgroups of BC to normal samples. They do not concentrate on any of the subtypes like basal like subtype. So the DE circRNAs were selected from the list of circRNAs DE between all subtypes compared to control samples like DE genes. But the DE miRNAs were selected from the list of DE miRNAs from the comparison of basal like subtype vs normal controls. This selection criteria could affect the network analysis since the elements (circRNA-miRNA-mRNA) of the network are selected from different subgroups.

The place of the Figure 3 is confusing, and it would be better to present it after Figure 5. Otherwise, it is not compatible with the flow of the text.

6. PLOS authors have the option to publish the peer review history of their article (what does this mean?). If published, this will include your full peer review and any attached files.

Reviewer #1: No

Reviewer #2: No

---

## [Author Response · Author response to Decision Letter 0]

22 Sep 2023

Dear Editor,

We thank the reviewers for the constructive and thoughtful comments and suggestions on our manuscript entitled “Bioinformatics Analysis of the Potentially Functional circRNA-miRNA-mRNA Network in Breast Cancer” (Manuscript Number: PONE-D-23-03451). We have revised the manuscript to address each of the comments. Below are our point-to-point responses to the reviewers’ comments. 

The modifications in the manuscript are highlighted within the “Revised-PONE-D-23-03451-with-track.docx” document. We also provided an unmarked version of the revised document (Revised-PONE-D-23-03451-unmarked.docx) upon your request.

Sincerely,

Zeyneb Kurt

Comments from the Reviewer #1 and the authors’ responses to the comments:

Comment 1: 

In this manuscript the authors aimed to find out regulatory networks of circRNAs, miRNAs and mRNAs in breast cancer. The manuscript has an importance since finding out circuits and networks is important to understand the development and progression of breast cancer from systems biology approach. The authors have to be more precise about the hypothesis they suggested; it can not be extracted from the manuscript whether they are dealing with the networks related to the formation (T/N) or the molecular classification (PAM50) of breast cancer. And the authors should consider the points raised below to be accepted;

Response 1: 

We appreciate the reviewer’s valuable comments. We have investigated the interactions among the circRNA-microRNA-mRNA for different molecular (i.e. PAM50) subtypes of breast cancer, rather than limiting our research to a T/N comparison-based investigation. The differentially expressed (DE) miRNA and mRNA results have been reported for each PAM50 subtype individually in the Sections 3.1.2 and 3.1.3, respectively. Further analysis including the molecular network construction and GO analysis have been performed for the DE miRNAs and mRNAs shared across all PAM50 subtypes with an emphasis on implications of our findings on the BLBC subtype due to its higher recurrency rate and poorer survival outcomes as also outlined in the final paragraph of this response section. Now, we have revised the manuscript and elaborated our main hypothesis and research questions better in Introduction (e.g. final paragraph, Lines 90-97 in the unmarked version) as well as in Discussion. 

We would also like to declare that the circRNA datasets available in the data repositories either do not contain information regarding the molecular subtypes of their samples at all (GSE182471) or only include two of the PAM50 molecular subtypes, i.e. Luminal A and TNBC (GSE101124). However, microRNA and mRNA datasets contain information for the samples from all PAM50 subtypes. Since our investigation starts from the DE mRNA and proceeds to the downstream stages through adversely mapping them to the DE miRNA and later on to the DE circRNAs, the molecular subtype knowledge of the samples preserved from the upstream towards the downstream our analysis. We have now clarified these points in Introduction. 

We would also like to note that TNBC has a high degree of overlap with the subtype BLBC and it was stated that most experts would tackle them as the same subtypes; however a small fraction of TNBC has been claimed to include non-basal-like (NBL or NBLBC) type as well. Hence TNBC is claimed to have two types, BLBC and NBLBC, with BLBC being the more aggressive one [https://doi.org/10.1155/2020/4061063].

In the GSE101124 dataset, TNBC versus normal tissue samples analysis was also performed, taking into account the referee's recommendation. Compared to the normal group, hsa_circRNA_100435 and hsa_circRNA_101004 were down-regulated in the TNBC group, while hsa_circRNA_000585 was up-regulated (p<0.05; logFC>1.5). These results may indicate that these circRNAs highlighted in the study may play an important role in BLBC.

The following sentence has been added to the Discussion section;

“Analysis of TNBC versus normal tissue samples on the GSE101124 dataset results may indicate that these three circRNAs highlighted in the study may play an important role in BLBC. Moreover, although the expression alteration of genes, targeted by miRNAs that sponged via selected circRNAs, was more prominent in the BLBC and Her2-positive subgroups, it was much more limited in the normal-like and Lum-A groups (an expression alteration was detected as similar to a pan flute, Fig. 6).”

A relevant response was also provided in Response 5 to Comment-5 from Reviewer 1.

Comment 2:

The authors need to revise the word “database” to data sets” in the following sentence, which is in the introduction section of the manuscript; “Therefore in our study circRNAs that may be relevant to BC were detected using the GSE101124 and GSE182471 databases.” These series are the datasets submitted to GEO database.

Response 2:

We thank the reviewer for their careful reading and comment. This term has now been updated accordingly (introduction, line 83 in the in the unmarked version).

Comment 3:

Although the main aim of the study is defined as to find out the regulative circRNA-miRNA-mRNA network in breast cancer, they do not mention anything about these networks or the structure of the network or the significance of finding out networks in the introduction. Introduction is too brief and not so informative at all.

Response 3: 

This is a valid point raised. Now, we described the structure and significance of network representations in relevant works and justified for using a network in our study. Introduction is extended accordingly. 

We have updated the introduction and added statements as follows:

“It has been noted that biological processes in living cells would better be modeled by networks since molecular phenotypes do not operate in isolation, instead their interactions collectively carry out these processes [4]. Hence, a network representation can provide a better understanding of the biological and molecular processes beyond analyzing a single molecule or gene, for instance identified from differentially expressed gene analyses. It is expected that identifying circRNA-miRNA-mRNA connections will be essential in explaining the molecular processes of numerous illnesses, detecting biomarkers for early diagnosis, and expanding therapy choices. A single miRNA has the capacity to target hundreds of genes, while a single circRNA can serve as a sponge for dozens of miRNAs. Using bioinformatics data to simplify the circRNA-miRNA-mRNA interactions, which are comprised of such complicated processes, can shed light on in vitro and in vivo investigations. For example, in the bioinformatics study of Liu et al [5] , it was emphasized that hsa_circRNA_0003638 may play a role in the pathogenesis of atrial fibrillation by targeting the CXCR4 gene via hsa-miR-1207-3p. Similarly, Hu et al [6] suggested that the interaction of hsa_circ_0009581/hsa-miR-150-5p, and hsa_circ_0001947/hsa-miR-454-3p may play a role in the AML cancer process.”

Comment 4:

The authors have to give the details of the selection criteria of the expression data of circRNAs. Why did they use only two array data instead of adding NGS data to increase the sample number. They may explain this as a limitation of the study.

Response 4: 

We thank the reviewer for their careful and helpful comment.

The GEO datasets were searched in the NCBI Geodataset database with the keywords "circRNA, breast; circular RNA, breast; circRNA, breast cancer; circular RNA, breast cancer". However, it has been determined that there are no NGS datasets studied in breast cancer tissues. It was observed that there were NGS datasets related to breast cancer cell lines, but the study was continued with GSE101124 and GSE182471 datasets since only tissue samples were included. 

Comment 5:

One of the data set GSE101124 is comprised of both breast cancer cell lines and breast tumors with different subtypes (Luminal A and TNBCs). And the subtypes of the tumor samples are not mentioned in the study of GSE182471. The authors need to explain the characteristics of the data they used in the manuscript rather than to give it up to the readers. As far as it is understood the authors analyzed the cell line data together with the tumor data which is completely irrelevant. They have to exclude the cell line data since their genetic makeup is completely different from the tumor samples as they are cultured, and many passages are performed.

Response 5:

The circRNA dataset GSE182471 does not include molecular subtype information regarding the samples (samples are described as either tumour or adjacent-non-tumour), meanwhile GSE101124 includes only two types (LumA and TNBC), as also depicted by the reviewer. Meanwhile miRNA and mRNA datasets do have samples from all available molecular subtypes. We have clarified that in the manuscript (Section 2.1.1), which has been also briefly touched on response #1 to comment #1 from the reviewer#1. 

We found the comment raised by the reviewer on the use of cell lines together with the breast tumours on point, useful and valid. We have removed the cell lines from our analyses and updated our findings accordingly (Section 3.1.1 summarise the novel findings upon this change). The number of the DECs shared between the two datasets is reduced from 16 to 13 (Section 3.1.1).

Comment 6:

As it can be extracted from the figure 2 but not from the text, the authors found out the differentially expressed circRNAs among all groups of tumors compared to control samples (tumors (LumA, TNBC, cell lines) vs controls) but although they found out DE miRNAs between all subtypes they only concentrated on the basal tumors and controls. Then they related these DE miRNAs and circRNAs, which are obtained from different sample types. The authors must explain the selection criteria of the samples to be compared clearly. Otherwise, the relationship they are trying to establish between circRNAs and miRNAs could not be robust.

Response 6:

We now clarified it better that our analyses and findings have included the DE miRNA and mRNA for all PAM50 subtypes, not only for the basal tumours (e.g. Sections 3.1.2 and 3.1.3 describe findings from all PAM50 subtypes). However due to the aggressiveness and poor survival outcome of the BLBC, our discussion and results emphasise the importance of our findings on this specific subtype (for example as shown in Figure 6 the greatest fold change was observed for the BLBC).

We also amended the methods section to describe our methodology better; which starts from the identification of DE mRNAs for each subtype, maps them to the DE miRNAs for each subtype (by considering the direction of the regulation) with using the knowledge on the binding miRNA-mRNA pairs from the mirDB, miRTarBase, and miRWalk databases; then mapped the narrowed down DE miRNAs to the DE circRNAs by considering the direction of the regulation and using the information from CSCD v2.0 database. Due to not having any circRNA resources regarding the PAM50 subtypes, we have used the DE circRNAs found from a tumour vs normal comparison, different from the DE analyses for miRNAs and mRNAs. But we hypothesised that, since we are coming from the upper stream of the circRNA-miRNA-mRNA axis, we can relate the DE molecules of different PAM50 subtypes to the potential BC-driving circRNAs. We clarified our hypothesis and descriptions in the Introduction accordingly.

We have reported the DEM and DEG findings from all PAM50 subtypes with an emphasis on how our findings inform especially the basal-like subtype which is the most aggressive and mortal subtype among all. We have listed the number of DEMs and DEGs for each subtype in the manuscript (Sections 3.1.2 and 3.1.3). DEGs were filtered out according to the criteria given in Section 2.3 and Figure 6 clearly evidences how the log2-transformed fold change (LogFC) values of the selected 18 DEGs change from BLBC to normal-like tumour almost in the form of a pan flute shape as described in Section 3.4. 

Comment 7:

The selection criteria of 3 DE circRNAs among 16 common DE circRNAs for further studies is not justified. The authors have to be more specific about the biological relevance of why they selected these three. And the same comment is valid for the selection of DE 3 miRNAs. They need to explain it in the results section more clearly.

Response 7:

The three DE circRNAs were shortlisted from 13 DECs (previously this was 16 but once the cell lines were removed from the dataset GSE101124, 13 DECs remained) shared between the two circRNA expression datasets, based on (i)the presence of the circRNA-miRNA pairs in the CSCD v2.0 database (ii) the direction of the regulation of the DE miRNAs and DE circRNAs, since they are expected to be inversely related. This filtering criteria have been now described in Section 2.5 upon the comment from the Reviewer#1. This has led us to have three circRNAs and matching three miRNAs (Section 3.3) in the end.

Comment 8:

As far as I understand the authors selected DE genes by comparing all subgroups of BC to normal samples. They do not concentrate on any of the subtypes like basal like subtype. So the DE circRNAs were selected from the list of circRNAs DE between all subtypes compared to control samples like DE genes. But the DE miRNAs were selected from the list of DE miRNAs from the comparison of basal like subtype vs normal controls. This selection criteria could affect the network analysis since the elements (circRNA-miRNA-mRNA) of the network are selected from different subgroups.

Response 8:

We have identified the DE genes for each PAM50 subtype individually, just like we did for the DE miRNAs since the subtype data was available for the samples for those molecular data types. Then we kept the DE genes and miRNAs that are shared across all PAM50 subtypes. However, for the circRNAs, we could not use subtype information since the circRNA dataset samples lack this knowledge. This has been mentioned in our response to Comment#6 from the Reviewer#1. Manuscript was edited to clarify this point (Sections 2.3, 2.5, 3.1 subsections).

Comment 9:

The place of the Figure 3 is confusing, and it would be better to present it after Figure 5. Otherwise, it is not compatible with the flow of the text.

Response 9:

Thank you to the reviewer for their helpful comments, now we moved the survival analysis to section 3.5 and the order of the figures have been updated; survival figures are presented at the end now (Figure 7 C-E).

Comments from the Reviewer #2 and the authors’ responses to the comments:

Circular RNAs (circRNA) are involved in regulating proliferation, invasion, and migration of cancer and may serve as novel molecular markers for diagnosis, prognosis evaluation and treatment of cancer. The current study represents a bioinformatic-based analysis to identify potential circRNA-based prognostic and targeted therapy strategies for Breast Cancer (BC). Here, the authors indicated that hsa_circ_0000515/ miR-486-5p/ SDC1 axis may be an important potential biomarker for BC patients in the basal-like breast cancer (BLBC) group. The overall level of the paper is good and important issues are emphasized. However, some issues listed below need to be addressed by the authors to enhance their manuscript.

Comment 1:

Authors are requested to care using full expression of some terms followed by its abbreviation in the paper.

Response 1:

We appreciate the reviewer’s valuable comments and summary.

Full expressions of the acronyms have been carefully completed in the manuscript now. 

The full text of the updated terms, followed by their abbreviations, are listed as follows:

Gene Expression Omnibus (GEO), circular RNAs (circRNAs), The Cancer Genome Atlas (TCGA), Molecular Taxonomy of Breast Cancer International Consortium (METABRIC), microRNAs (miRNAs), Gene Ontology (GO), Kyoto Encyclopedia of Genes and Genomes (KEGG).

Comment 2:

To identify differentially expressed circRNAs (DECs), Authors downloaded two datasets containing circRNA expression profiles in BC patients from Gene Expression Omnibus (GEO) database. Authors are requested to give related links or references for databases or other web-based tools (for example: CSCD v2.0) that are used in the bioinformatic analysis in the materials and method section.

Response 2:

We thank the reviewer for their careful reading, indeed the links should have been provided. We now added the links and references to the databases and web tools that provide us the information regarding the molecular interactions (Sections 2.1.1 and 2.5). Moreover, details of the databases, tools, and R packages used in this study are also provided in S1 Table.

Comment 3:

The cutoff parameters should be given for STRING analysis in the “Analysis of the protein–protein interaction (PPI) network” title of the materials and methods section.

Response 3:

We now described the selected parameters for the PPI network in Methods Section 2.7.

Comment 4:

The authors found differentially expressed circRNAs (DECs) by using two independent GEO datasets (GSE101124 and GSE182471). This analysis was conducted in Breast Cancer Compared to Adjacent Tissues. However, Identification of Differentially Expressed mRNAs and miRNA were performed in PAM50 breast cancer subtypes using TCGA and metabric datasets. Authors shared the mRNA and miRNA analysis for BLCL subtype. Each of the five breast cancer molecular subtypes vary by their biological properties and prognoses. The circRNAs that were found in the current study might have a different profile when we compared the expression profile as subtype specific. The Authors are asked why they didn’t compare TCGA and metabric datasets as tumor vs normal and whether this article belongs to breast cancer in general or to the BLCL subtype. Authors are requested to explain this issue.

Response 4:

We thank the reviewer for raising this point. In our study design, we started from the identification of DE mRNAs for each PAM50 subtype, then mapped them to the DE miRNAs, (that are found for each PAM50 subtype) by considering the direction of the regulation of DE genes and miRNAs as well as by using the knowledge on the matching pairs of miRNA-mRNA from the mirDB, miRTarBase, and miRWalk databases. After that, we mapped the shortlisted DE miRNAs to the DE circRNAs by considering the direction of the regulation and using the information from the CSCD v2.0 database (now this has been detailed further in Section 2.5). As also depicted by the reviewer, there were no circRNA resources regarding the PAM50 subtypes and we predicted the DE circRNAs from a comparison made on tumour vs normal, which was different from the DE analyses for miRNAs and mRNAs. However, only one of the circRNA datasets we analysed, GSE101124, has the subtypes LumA and TNBC (although it does not include all PAM50 subtypes), as well as normal tissue samples. We performed TNBC versus normal tissue samples analysis (which has been also provided as a response to the Comment#1 from the Reviewer#1). Compared to the normal group, hsa_circRNA_100435 and hsa_circRNA_101004 were down-regulated in the TNBC group, while hsa_circRNA_000585 was up-regulated (p<0.05; logFC>1.5). These results may indicate that these circRNAs highlighted in the study may play an important role in BLBC.

We hypothesised that, since we are coming from the upper stream of the circRNA-miRNA-mRNA axis, we can relate the DE mRNAs and later on the DE miRNAs of different PAM50 subtypes to the candidate BC-driving circRNAs. Now, we described this study design in the manuscript more explicitly (e.g. in Introduction, Lines 90-97 in the unmarked version). We also clarified the selection criteria of the molecules in Sections 2.3 and 2.5. This amendment has been also communicated in another response to the comments from the other reviewer, i.e. Reviewer#1 (Comments & response #6).

Our manuscript reports the findings for all PAM50 subtypes but due to the aggressiveness and poor survival outcome of the BLBC, our discussion emphasises our findings’ implications on the BLBC subtype further despite we have presented the findings from all PAM50 subtypes that are available in results Sections 3.1.2 and 3.1.3.

Comment 5:

There is a similar study in the literature which was published in 2021 (https://doi.org/10.1155/2021/1732176). This study could affect the novelty of this manuscript. The authors are requested to discuss this study in terms of its advantages and disadvantages in the discussion section.

Response 5:

We thank the reviewer for raising this point. Indeed, this study has similarity to our work, we have now discussed and highlighted differences between this study and ours, and clarified the novelty of our work including (i) investigation of circRNA-miRNA-mRNA axes for different PAM50 subtypes rather than conducting a binary model (i.e. tumour vs control), which has not been investigated previously; (ii) use of multiple datasets (e.g two circRNA microarray, two miRNA and one mRNA expression datasets) meanwhile the referred study has used only one dataset per molecule type. This study has been cited and differences in our study design has been described in introduction and discussion sections. 

Comment 6:

Calculation of Spearman’s rank correlation coefficient between DE miRNA and DE circRNA/mRNA expression profiles could be better.

Response 6:

This is a brilliant recommendation. Now, we added a correlation-based analysis (using Spearman correlation) for the DE mRNAs and miRNAs predicted from our analyses. We selected the common samples from miRNA and mRNA datasets to estimate the correlation scores between the shortlisted miRNAs and mRNAs using the TCGA dataset and showed the correlation patterns and the relevant heatmaps in Figure 7A-B (Section 2.6). Figure 7 represents the correlation patterns (A) across all samples and (B) across solely BLBC-subtype samples. 

We are unable to add the correlation patterns between the circRNA and miRNA or mRNAs because there is no circRNA data exist from the matching miRNA and mRNA samples. 

Comment 7:

According to the ceRNA hypothesis, circRNA can act as effective ceRNAs depending on their abundance and cytoplasmic localization. Although circRNAs are enriched mainly in the cytoplasm, some studies on circRNAs indicate different subcellular localization. Therefore, subcellular localization analysis might be important for the hsa_circ_0000515/miR-486-5p/SDC1 axis. To support this axis, the authors can at least use a prediction tool such as Circ-LocNet for subcellular localization and coding potential analysis.

Response 7:

We thank the reviewer for their valuable comments. We tried to register the Circ-LocNet server (available on https://circ_rna_location_predictor.opendfki.de/), three co-authors have tried that but we could not get any user credentials that would enable us to use this server in ‘training mode’. Then, we attempted using the ‘prediction mode’ option since it has been available/open to guest users and does not require any registration. We have uploaded the sequence of our three circRNA; prediction mode has estimated class indices but we were not provided any further details, whether this prediction was made upon a previously trained model on the system (which might be assigned as a default classifier/predictor model with default parameters and trained on a dataset by the developers of that web server). We have checked the tutorial link on the server but could find any explanation regarding what the generated/predicted class labels are referring to for our shortlisted three circRNAs. 

The circRNAs hsa_circ_0016201 and hsa_circ_0000515 were predicted to be from the class-1 and hsa_circ_0000375 was estimated to be from class-7 but we are not provided information regarding the class labels and their matching class names, or cellular locations (this information was not present in the relevant publication either). Hence, we could not make any conclusion from that server and are not able to describe the models or parameters used in the prediction mode (to explain those to our readers in our methods section). We were also unable to interpret the outcome predicted by the server, as described earlier.

Comment 8: 

This study needs validation such as qRT-PCR for the potential candidates.

Response 8:

We thank the author for their valuable comment on the validation. However, we do not have the proper number of PAM50 breast cancer "tumor and adjacent normal tissue" to conduct a qRT-PCR experiment on the shortlisted molecules. While designing this bioinformatics study, in addition to the absence of tissue samples, we have not had any budget for data collection and funding for any functional validation of the predicted circRNAs, miRNAs, and mRNAs. We have recently earned a grant from the Research Universities Support Program (Project ID: TSA-2023-39483 and Project Title: Investigation of Specific circRNA/miRNA/Target Gene Axis in Breast Cancer Subtypes), long after we have completed and submitted the current manuscript. Within the scope of our accepted project, PAM50 subtypes of breast cancer "tumor and adjacent normal tissue" samples have started to be collected only very recently. It is predicted that it may take approximately 1-1.5 years for the samples to reach sufficient numbers for statistical evaluation in expression analysis. Briefly, we could only use the in-silico validation algorithm, as we currently do not have enough tissue samples to perform qRT-PCR testing on our candidate circRNA/miRNA/mRNA axis. However, in future work, we are willing to use our methodology proposed here and conduct functional experiments on our findings, for which we do not have proper resources (in terms of data) currently. Although there are a few studies [1] in the literature on the relationship between circRNA-breast cancer subtypes using breast cancer cell line datasets, the information on this area is still quite insufficient.

This study may highlight the lack of studies in the literature comparing circRNA, miRNA and target gene expression differences in BC subgroups. In addition, our current study, despite the lack of qRT-PCR validation, has the potential to contribute to an increase in studies for a better understanding of BC subgroups. Therefore, we hope that the esteemed referee will understand the situation.

Reference:

[1] Huang Y, Qian M, Chu J, Chen L, Jian W, Wang G. Identification of circRNA-miRNA-mRNA network in luminal breast cancers by integrated analysis of microarray datasets. Front Mol Biosci. 2023 Apr 28;10:1162259. doi: 10.3389/fmolb.2023.1162259. PMID: 37187897; PMCID: PMC10175596.

Comment 9:

Similar to DEM and DEC, DEG results should also be represented in the fig 2.

Response 9:

Since there is only a single dataset was used for the DEG analysis, TCGA (as the METABRIC dataset does not have normal samples to enable comparison for a DEG analysis), we are unable to show a pair-wised Venn diagram showing the common DEGs in multiple datasets. However we can show a Venn diagram of DEGs across different PAM50 subtypes (added as Figure 2G-H).

Comment 10:

Authors expressed the 16 DECs as the most significantly changed ciRCRNAs in the “2.2 Predicting the associated biological features” part of the results section. 16 circRNAs, overlapping circRNAs from 2 GEO circRNA datasets or the most significantly changing circRNA?

Response 10:

We thank the reviewer for their careful reading; there are DECs overlapping between the two circRNA datasets; now we have amended the text to describe this explicitly (Section 3.1.1). Currently, the number of the overlapping DECs has decreased to 13 because we have removed four cell lines from the GSE101124 dataset upon a valid point raised by the Reviewer#1 (their comment#5), removal of the cell lines has led to a change in the list of identified DECs. 

Comment 11:

The selection strategy of the 3 candidate circRNA (has_circRNA_0000375, has_circRNA_0000515, has_circRNA_0016201) is not clear. Authors are requested to explain how to find these 3 circRNAs? Selection of 3 miRNAs is also unclear. Selection of the miRNA and mRNAs could be explained in the results according to Section 2.2 in the materials and methods in detail. Authors indicated that 11 common miRNAs from TCGA and metabric miRNA data sets analysis. Selection of 3 miRNAs, 3 circRNAs and genes could be shared in a better way.

Response 11:

We have checked the existence of the DECs, DEMs and DEGs in the databases (mirDB, miRTarBase, and miRWalk databases for the mRNA-miRNA pairs; CSCD (v2.0) database for the miRNA-circRNA pairs). We have also considered the direction of the regulation (e.g. if a DEM binds to a specific DEC, one of this pair should be upregulated while the other one downregulated). Three DECs among 13 shortlisted are present in the CSCD database and they are mapping to three miRNA from our shortlisted DEM list. This process has now been described further in Section 2.5 and results have been now described better in Sections 3.3 and 3.4. This has been also explained in Response #4 to the Comment #4 from the reviewer (R#2); as well as Response and comment #7 from the Reviewer#1. 

Comment 12:

Metabric dataset also has mRNA expression analysis data. Authors could have also used this dataset for DEG analysis or for validation.

Response 12:

Thank you for the comment provided. Unfortunately the METABRIC dataset does not include control (healthy) samples in the mRNA expression dataset, but include data only from the pam50 subtypes. Hence, comparison of a subtype vs control (or overall, tumour vs control comparison) is not applicable to the METABRIC’s mRNA dataset. This has been described and clarified in the main text now (Section 2.1.2).

Comment 13:

Authors have found 18 genes from TCGA Breast cancer mRNA data analysis according to their criteria. However, they showed the survival analysis results for four of them. It is not clear why only these four genes were represented.

Response 13:

We now show the survival analysis of all selected 18 DEG genes that are shared across all PAM50 subtypes in the S5 Fig and S6 Fig for overall and disease-free survival, respectively (in the TCGA dataset). Among them, four genes showed a significant difference in survival (so were shown in the main text), meanwhile the rest have not shown a significant difference in survival analysis.

---

## [Decision Letter · Decision Letter 1]

26 Mar 2024

Bioinformatics Analysis of the Potentially Functional circRNA-miRNA-mRNA Network in Breast Cancer

PONE-D-23-03451R1

Dear Dr. Kurt,

We’re pleased to inform you that your manuscript has been judged scientifically suitable for publication and will be formally accepted for publication once it meets all outstanding technical requirements.

Kind regards,

Jianhong Zhou

Staff Editor

PLOS ONE

Additional Editor Comments (optional):

Reviewers' comments:

Reviewer's Responses to Questions

**Comments to the Author**

1. If the authors have adequately addressed your comments raised in a previous round of review and you feel that this manuscript is now acceptable for publication, you may indicate that here to bypass the “Comments to the Author” section, enter your conflict of interest statement in the “Confidential to Editor” section, and submit your "Accept" recommendation.

Reviewer #1: All comments have been addressed

2. Is the manuscript technically sound, and do the data support the conclusions?

Reviewer #1: Yes

3. Has the statistical analysis been performed appropriately and rigorously? 

Reviewer #1: Yes

4. Have the authors made all data underlying the findings in their manuscript fully available?

Reviewer #1: Yes

5. Is the manuscript presented in an intelligible fashion and written in standard English?

Reviewer #1: Yes

6. Review Comments to the Author

Reviewer #1: The authors have satisfactory improved their paper, in reaction to the comments. I would like to thank the authors for their great efforts.

7. PLOS authors have the option to publish the peer review history of their article (what does this mean?). If published, this will include your full peer review and any attached files.

Reviewer #1: No

---

## [Editor Report · Acceptance letter]

4 Apr 2024

PONE-D-23-03451R1 

PLOS ONE

Dear Dr. Kurt, 

I'm pleased to inform you that your manuscript has been deemed suitable for publication in PLOS ONE. Congratulations! Your manuscript is now being handed over to our production team.

Kind regards, 

on behalf of

Dr. Jianhong Zhou 

Staff Editor

PLOS ONE